# Isotopic signatures induced by upwelling reveal regional fish stocks in Lake Tanganyika

**Benedikt Ehrenfels**[1,2‡*], **Julian Junker**[3,4‡], **Demmy Namutebi**[4,5], **Cameron M. Callbeck**[1], **Christian Dinkel**[1], **Anthony Kalangali**[6], **Ismael A. Kimirei**[6,7], **Athanasio S. Mbonde**[6], **Julieth B. Mosille**[6], **Emmanuel A. Sweke**[6,8], **Carsten J. Schubert**[1,2], **Ole Seehausen**[3,4], **Catherine E. Wagner**[9], **Bernhard Wehrli**[1,2]

1 Eawag, Swiss Federal Institute of Aquatic Science and Technology, Department Surface Waters – Research and Management, Kastanienbaum, Switzerland, 2 ETH Zurich, Institute of Biogeochemistry and Pollutant Dynamics, Zurich, Switzerland, 3 Eawag, Swiss Federal Institute of Aquatic Science and Technology, Department Fish Ecology and Evolution, Kastanienbaum, Switzerland, 4 Institute of Ecology & Evolution, University of Bern, Bern, Switzerland, 5 IHE Delft, Institute for Water Education, Delft, Netherlands, 6 TAFIRI, Tanzania Fisheries Research Institute, Kigoma, Tanzania, 7 TAFIRI, Tanzania Fisheries Research Institute, Dar es Salaam, Tanzania, 8 DSFA, Deep Sea Fishing Authority, Zanzibar, Tanzania, 9 Department of Botany and Program in Ecology, University of Wyoming, Laramie, Wyoming, United States of America

‡ BE and JJ share first authorship on this work.
* benedikt.ehrenfels@eawag.ch

**Data Availability Statement:** The data sets are uploaded to an open access repository: https://doi. org/10.3929/ethz-b-000600742. Other related and previously published data can be found here: https://doi.org/10.3929/ethz-b-000418479.

## Abstract

Lake Tanganyika's pelagic fish sustain the second largest inland fishery in Africa and are under pressure from heavy fishing and global warming related increases in stratification. The strength of water column stratification varies regionally, with a more stratified north and an upwelling-driven, biologically more productive south. Only little is known about whether such regional hydrodynamic regimes induce ecological or genetic differences among populations of highly mobile, pelagic fish inhabiting these different areas. Here, we examine whether the regional contrasts leave distinct isotopic imprints in the pelagic fish of Lake Tanganyika, which may reveal differences in diet or lipid content. We conducted two lake-wide campaigns during different seasons and collected physical, nutrient, chlorophyll, phytoplankton and zooplankton data. Additionally, we analyzed the pelagic fish–the clupeids *Stolothrissa tanganicae*, *Limnothrissa miodon* and four *Lates* species–for their isotopic and elemental carbon (C) and nitrogen (N) compositions. The $\delta^{13}C$ values were significantly higher in the productive south after the upwelling/mixing period across all trophic levels, implying that the fish have regional foraging grounds, and thus record these latitudinal isotope gradients. By combining our isotope data with previous genetic results showing little geographic structure, we demonstrate that the fish reside in a region for a season or longer. Between specimens from the north and south we found no strong evidence for varying trophic levels or lipid contents, based on their bulk $\delta^{15}N$ and C:N ratios. We suggest that the development of regional trophic or physiological differences may be inhibited by the lake-wide gene flow on the long term. Overall, our findings show that the pelagic fish species, despite not showing evidence for genetic structure at the basin scale, form regional stocks at the seasonal timescales. This implies that sustainable management strategies may consider adopting regional fishing quotas.

**Funding:** This work was funded by the Swiss National Science Foundation. The grant CR23I2-166589 for the project titled "From biogeochemistry to the ecological genomics of pelagic fish stocks - a study across 4 trophic levels" was awarded to Bernhard Wehrli and Ole Seehausen (https://data.snf.ch/grants/grant/166589). The funders had no role in study design, data collection and analysis, decision to publish, or preparation of the manuscript.

**Competing interests:** The authors have declared that no competing interests exist.

## 1. Introduction

Lake Tanganyika is by volume the second largest freshwater lake in the world, and its pelagic fish community sustains the second largest inland fishery in Africa [1], providing important employment opportunities and animal protein for millions of people in the riparian communities [2, 3]. The pelagic food web in Lake Tanganyika is composed of a copepod-dominated zooplankton assemblage, a phyto- and zooplankton grazer community consisting of two endemic sardine species (*Stolothrissa tanganicae* and *Limnothrissa miodon*), and a predator assemblage comprising of four endemic latid species (genus *Lates*), of which *Lates stappersii* is the most common [4]. *Stolothrissa* juveniles mainly feed on phytoplankton, particularly diatoms, whereas adults prefer zooplankton (Fig 1; [4]). The larger sardine, *Limnothrissa*, also feeds on phytoplankton as juvenile; the share of zooplankton increases with size, whereby large adults also prey upon small *Stolothrissa* [4, 5]. *Stolothrissa* are also a major prey of adult *Lates stappersii*, which supplement their diet with zooplankton [6]. *Stolothrissa* and *Lates stappersii* spawn offshore, whereas *Limnothrissa* and the three larger *Lates* species, *L. microlepis*, *L. mariae*, and *L. angustifrons* nearshore [4, 7, 8]. Today, the sardines and *Lates stappersii* account for 95% of the pelagic fish catch in Lake Tanganyika [9].

The pelagic fish stocks suffer from heavy fishing [2, 10] and from a long term decline that was attributed to climate change [11–13]. The increased warming of the surface waters caused by climate change leads to steep temperature gradients in the water column. These gradients build physical barriers to vertical mixing, thereby limiting the transfer of nutrients to surface waters where light is available to drive primary productivity [11, 12, 14–17].

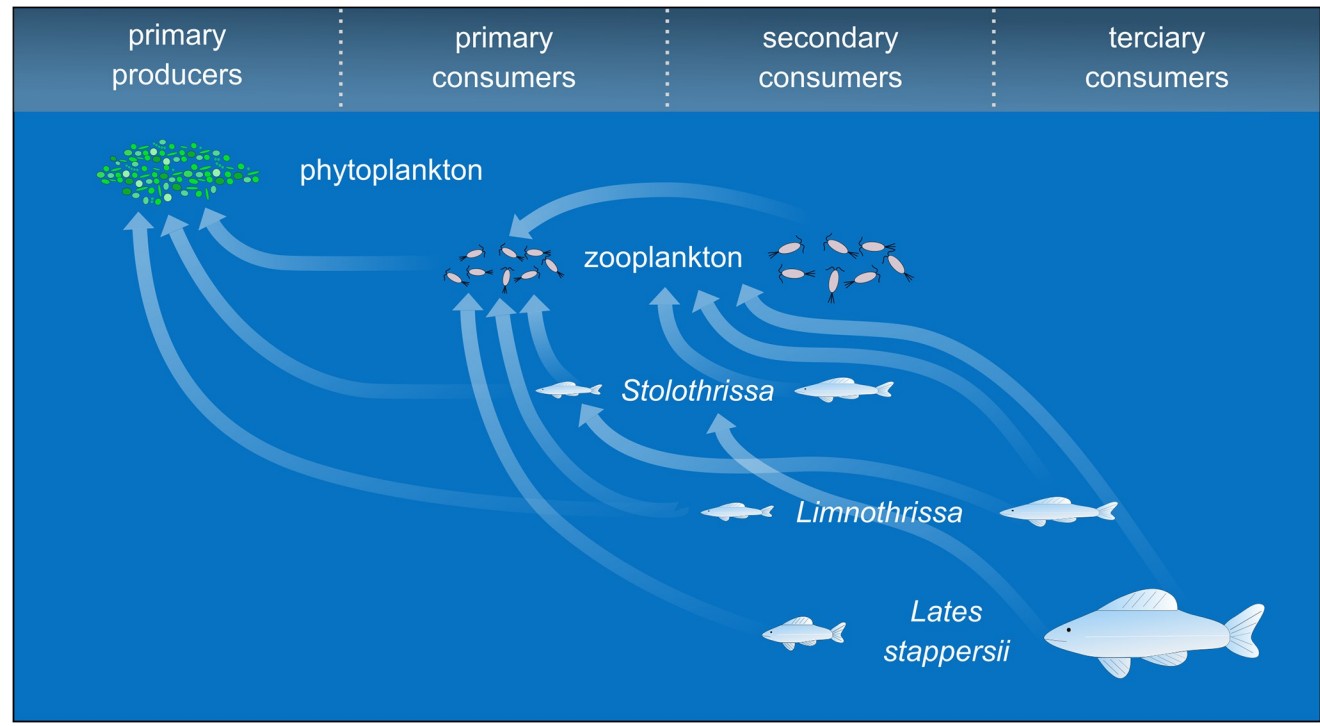

**Fig 1. Simplified schematic of the pelagic food web of Lake Tanganyika.** Zooplankton are depicted as primary or secondary consumers and the three fish species are shown as juveniles or adults. Arrows indicate major predator-prey relationships. These depictions are coarse, and the precise location of the arrow tip does not necessarily indicate a preference for zooplankton higher or lower in the food web. The relative sizes of the food web members are not to scale.

Assessing potential long-term changes in the ecology of the pelagic fish is impaired by data scarcity, but Lake Tanganyika's limnological cycle offers the opportunity to study the impact of varying levels of stratification on a basin-scale. This annual cycle is driven by climatic differences between the north and south and is characterized by four stages (Fig 2a–2d; [18, 19]) (Fig 2a–2d): (i) In the warm rainy season (November-March), stagnant and highly stratified waters lead to an overall nutrient-depleted epilimnion (Fig 2a). (ii) In March-May, the southeast trade winds initiate the lake circulation in the upper water column, resulting in strong nutrient upwelling in the southern basin (Fig 2b). (iii) The upwelling in the south transforms into a convective mixing of the upper ~150 m. The sinking, cool surface waters in the south reverse the lake circulation by initiating a northward current between 50–100 m and a surface counter current, further weakening thermal stratification across the lake (Fig 2c). (iv) The

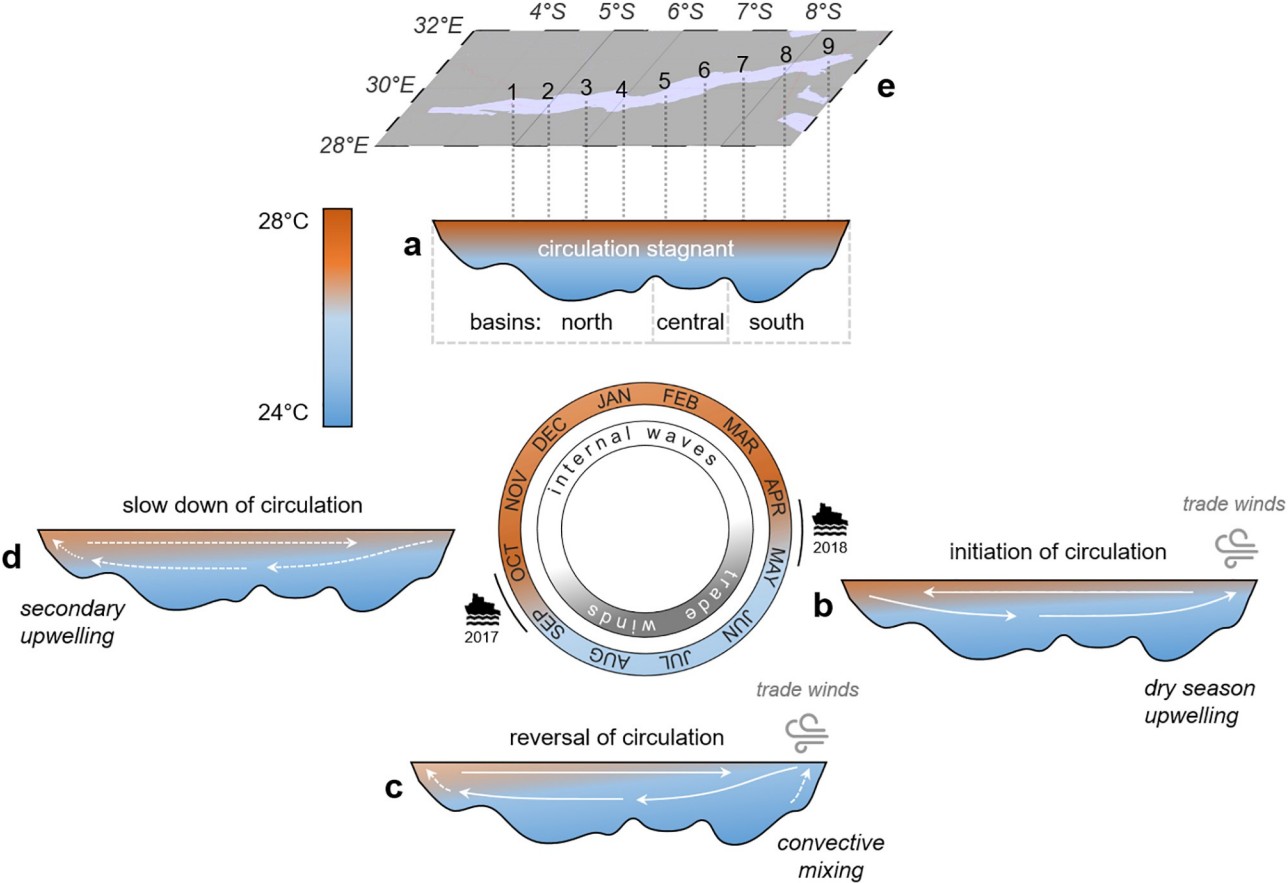

**Fig 2. The limnological cycle of Lake Tanganyika with its four major phases according to Plisnier et al. [18] and Verburg et al. [19].** (a) Stagnant, highly stratified waters during the warm rainy season (November-March) only support low nutrient availability. (b) The onset of the cool dry winds in March-May initiates the upwelling in the south leading to high nutrient fluxes in this region. (c) The lake circulation reverses during the dry season (May-September). Water column stratification is low and the nutrient availability high across the lake, with a maximum in the convective mixing area in the south. (d) The trade winds cease in October slowing down the lake circulation, while the water column re-stratifies. A weaker secondary upwelling leads to a nutrient pulse at the northern end of the lake. During the dry season, wind-driven upwelling and mixing are the dominant driving force behind nutrient injections into the euphotic zone, whereas internal waves are particularly important in the rainy season. The color gradient indicates the level of thermal stratification. Note that this latitudinal cross-section is not to scale and that the outlined mechanism primarily affects the upper water column (<200 m). Our two sampling campaigns were timed at the seasonal transitions in September/October and April/May to compare the effects of the preceding dry and rainy seasons. (e) The map shows the nine stations for water column and plankton sampling. Fish samples representing the pelagic catch were collected from the respective coastal villages/towns.

trade winds cease in October, slowing down the circulation, while the water column re-stratifies lake-wide (Fig 2d).

The limnological cycle of Lake Tanganyika leads to regional (i.e. basin-scale) and seasonal gradients in nutrient availability, causing overall higher primary productivity in the south than in the north of the lake and higher primary productivity in the dry season compared to the rainy season [20–24]. Besides primary productivity rates, the nutrient availability also affects the phytoplankton community composition, with blooms of highly nutritious diatoms occurring in the dry season [24]. High densities of zooplankton as well as the planktivorous sardines *Stolothrissa* and *Limnothrissa* are coupled to these phytoplankton blooms [7, 25, 26], which also seem to benefit sardine spawning and recruitment [27]. The north-south variability furthermore affects the zooplankton community composition: shrimps and calanoid copepods prevail in the south [28–30], whereas cyclopoid copepods and jellyfish dominate in the north [26, 31, 32]. Differences in the zooplankton community may in turn influence predatory fish. Mannini et al. [6] found that the diet of *Lates stappersii* in the north is heterogeneous and consists of copepods, shrimps, and sardines, whereas *Lates stappersii* in the south feed mainly on shrimps.

The regional variability in Lake Tanganyika's pelagic habitat, driven by the mixing regime, could additionally impact the life cycle (e.g. spawning phenology, developmental timing and recruitment success) of the pelagic fish species [8, 25, 27, 33]. The regional variation in the pelagic environment might generate different fitness optima and drive divergent adaptation of pelagic fish populations between north and south, if migration remains limited. However, recent genetic studies of the sardines [34, 35] and the four *Lates* species [36] did not find evidence for genetic population differentiation along the north-south gradient, suggesting that gene flow may overcome barriers for divergent natural selection between the basins, if they exist. Nonetheless, the fish stocks may still respond to regional differences in physicochemical conditions and food supply. Such ecological responses might involve variations in food web interactions [37] or lipid storage for bridging lean periods [38, 39]. However, effects of the regional variation and seasonality in the physical mixing regime of Lake Tanganyika on the distribution and ecology of its pelagic fish have not been studied.

Characterizing regional and seasonal gradients in the carbon (C) and nitrogen (N) elemental and isotopic composition of Lake Tanganyika's food web may provide insight to the migration distances, trophic levels, and lipid contents of the fish species. Marine studies demonstrate that the tissue of pelagic fish [40] and birds [41, 42] reflects the C and N isotopic signatures of the water masses they reside in. Thus, these isotopic markers can be used to assess the large-scale migratory and residency patterns of such pelagic animals. In this isotopic framework, the $^{13}C/^{12}C$ ratio or $\delta^{13}C$ increases only little from one trophic level to the next and therefore reflects the source of primary production [43–45]. Differences in primary productivity can alter the $\delta^{13}C$ of particulate organic matter (POM): high primary productivity results typically in high values of $\delta^{13}C$, due to the ongoing depletion of the of the DIC pool and decreasing discrimination against $^{13}C$ by phytoplankton [46–51]. Seasonal changes in $\delta^{13}C$ at the base of the food web can be tracked across trophic levels from plankton to fish in lake ecosystems [52]. In Lake Tanganyika for instance, O'Reilly et al. [11] and Verburg [16] used the $\delta^{13}C$-POM to reconstruct historical changes in primary productivity. In addition, previous $\delta^{13}C$ analyses in the northern basin have reported higher $\delta^{13}C$-POM values in the productive dry season compared to the rainy season [53–55]. The ratio of $^{15}N/^{14}N$, or $\delta^{15}N$, provides insight into the trophic position of an organism, because it increases significantly with each trophic level. This successive enrichment allows estimating an organism's trophic position in the food web and is used in ecology to describe prey and predator relationships [43–45]. Lastly, the elemental C:N ratio was often used to infer the lipid content of fish muscle tissue, with higher C:N denoting higher lipid contents [56, 57].

In this study, we explore the latitudinal and seasonal patterns of $\delta^{13}C$ and $\delta^{15}N$ in the pelagic food web of Lake Tanganyika in the context of the lake's limnological variability. During two lake-wide field campaigns in the final phases of the dry and the rainy seasons, we measured the C and N isotopic and elemental compositions of the major pelagic food web members (POM, zooplankton, the bivalve *Pleiodon spekii*, fish). These samples were collected in concert with limnological data including physical properties of the water column, oxygen and nutrient concentrations, chlorophyll, as well as the phyto- and zooplankton community and abundance [58]. Using the extensive data sets from those two contrasting time points, we first tested to what extent the regional and seasonal patterns in primary productivity induce systematic differences in the isotopic signatures of plankton, and then tracked the isotopic signals and C:N ratios through the food web to the pelagic fish. The results allowed us to assess the extent of regional isolation and ecological differentiation of the pelagic fish stocks. Assuming fish were regionally constrained within the investigated time period, instead of moving randomly across the lake, we expect fish to reflect the isotope signatures from the planktonic base of the food web. Thereby, when we refer to "regional", we imply that individuals move with a geographic reach equal or smaller than the basin-scale. Finally, we tested whether existing genetic differences [35, 36] were linked to dietary differences in the six major pelagic fish species.

## 2. Materials and methods

### 2.1 Study site and sampling

Our two Lake Tanganyika sampling campaigns, spanning two different hydrological conditions across a north-south transect of ~500 km, were conducted at the end of the dry season (28 September—8 October 2017) and the end of the following rainy season (27 April—7 May 2018). Water column and plankton characteristics were sampled during two cruises on *M/V Maman Benita* [58, 59]. At the end of the dry season, we collected fish samples as described in Junker et al. [35] at station 1, station 2, station 5, station 7 and station 9 during a land-based excursion prior to the cruise (17–24 September 2017), whereas all nine landing sites, corresponding to our nine pelagic sampling stations, were sampled during the cruise at the end of the rainy season (Fig 2e). In addition, we took fish samples in Kigoma in July 2017 [35].

### 2.2 Physical and chemical parameters

We measured temperature, dissolved oxygen, photosynthetically active radiation, and in-situ chlorophyll fluorescence via CTD profiling (Sea-Bird SBE 19plus) at each station. From these stations, we also collected water with large Niskin bottles (20–30 L) at 5–25 m depth intervals down to 250 m depth. Water column stratification was expressed as buoyancy frequency ($N^2$) and Schmidt stability (Sc). We interpreted clear peaks in $N^2$ as thermoclines, whereby the $N^2$ value at the peak provides a measure of steepness of the thermocline. In addition, we calculated the Sc over 1 $m^2$ between 50 and 100 m for each station using the R package 'rLakeAnalyzer' [60]. This depth interval extends from the typical location of the nitrate peak to the bottom of the euphotic zone [58, 61, 62]. For a more detailed description of the thermal structure of the water column see Ehrenfels et al. [58].

Water samples to measure nutrients (phosphate, ammonium, nitrate, and nitrite) were taken directly from the Niskin bottles, filtered sterile through 0.2 μm filters and processed onboard following standard methods [63–65]. On average, the detection limits were 0.22, 0.34, 0.20, and 0.03 μM for phosphate, ammonium, nitrate, and nitrite, respectively.

Water samples to measure dissolved inorganic carbon (DIC) were collected in 12 mL exetainers directly from the Niskin bottles and filtered sterile (0.2 μm). Samples were stored at

room temperatures and shipped to Switzerland. At Eawag Kastanienbaum, the DIC concentrations were measured by high temperature combustion catalytic oxidation using a Shimadzu TOC-L Analyzer (Shimadzu TOC-VCPH/CPN). A 2 mL aliquot of the DIC sample was used to quantify the isotopic fractionation of $\delta^{13}$C-DIC. The aliquot subsample was transferred to a new 12 mL exetainer, where it was Helium purged for 2 minutes. The sample was then capped and 50 μL orthophosphorous acid (85%) was added. The samples were mixed and stored for ~15 h at room temperature for equilibration prior to analysis by GC-IRMS (Isoprime). Sample $\delta^{13}$C-DIC were calibrated to the *Carrara marble* standard (ETH Zurich).

## 2.3 CO2 fixation rates

Carbon fixation incubations and rate calculations were done as described in Schunck et al. [66] and Callbeck et al. [67]. Briefly, samples were carefully filled from the Niskin into 4.5 L polycarbonate bottles capped with polypropylene membranes. Per sampled depth, we filled off triplicate bottles, including one control (no added label) and duplicate treatments (with amended $^{13}$C-HCO$_3^-$). We added 4.5 mL of $^{13}$C-bicarbonate solution (1 g $^{13}$C-bicarbonate in 50 ml water; Sigma Aldrich) to each of the treatment bottles. The label was mixed in the treatment bottles for ~30 min under shaking. Thereafter, a 12 mL subsample was taken for quantifying the labelling percent (mean 2.8%). The resulting headspace was re-filled with water from the same depth, and bottles were then incubated headspace-free in 60 L incubators covered with shaded light filters (LEE Filters) mimicking the in-situ irradiance and light spectrum. After 24 h, the samples were filtered on pre-combusted GF/F filters (Whatman). The filters were oven-dried (60°C for 48 h) and stored at ambient temperatures. Filter samples were shipped to Switzerland and further processed as described in 2.7. Due to the small difference between the in-situ and incubation temperatures (<5°C), the derived CO$_2$ fixation rates were not adjusted for temperature.

## 2.4 Chlorophyll, phytoplankton and particulate matter

We measured the chlorophyll-*a* concentrations according to Wasmund, Topp & Schories [68]. Briefly, 2–4 L of lake water were filtered through 47 mm glass fibre filters (GF55, Hahnemühle), which were directly transferred to 15 mL plastic tubes. Five mL ethanol (>90%) were added to the samples, followed by 10 min cold ultrasonification. The samples were stored at 5 °C overnight and sterile-filtered (0.2 μm) the following morning. The extracts were measured on-board with a fluorometer (Turner Trilogy) and calibrated against a chlorophyll-*a* standard (Lot# BCBS3622S, Sigma-Aldrich). Samples and standards were always handled and processed in the dark. In-situ chlorophyll fluorescence was calibrated against extracted chlorophyll-*a* samples and then used to calculate depth-integrated chlorophyll-*a* stocks (0–125 m).

For estimating the phytoplankton abundances, 4–10 L of water were concentrated to 20 mL using a 10 μm plankton net and fixed with alkaline Lugol solution. At TAFIRI Kigoma, phytoplankton cells were counted from 2 mL subsamples by inverted microscopy (at ×400 magnification). For particulate organic matter (POM), 2–4 L lake water was filtered through precombusted GF/F filters (nominal pore size 0.7 μm; Whatman).

## 2.5 Zooplankton and *Pleiodon spekii*

Zooplankton was collected with vertical net hauls across the oxygenated water column (0–150 m) at each pelagic station. We sampled different size fractions of the zooplankton community using three different nets. For smaller zooplankton, we used 25 and 95 μm nets with 0.03 and 0.02 m$^2$ mouth openings, respectively. A 250 μm net with a 0.28 m$^2$ mouth opening was used for larger, fast swimming species. We preserved all zooplankton collected from the first haul in

ethanol for taxonomic zooplankton community assessment, while the individuals from the second haul were designated for stable isotope analysis (only for samples from the 95 and 250 μm nets). At TAFIRI Kigoma, we analyzed the zooplankton community composition of the ethanol-preserved samples by compound microscopy (Leica Wild M3B) at x200 magnification. Additionally, we picked living individuals of the long-lived, filter-feeding bivalve *Pleiodon spekii* at near-shore habitats in water depths of 1.5–6 m by snorkelling. Bivalves were first euthanized with an overdose of MS222. Then we sampled the foot using clean scalpels and forceps and removed the mucous with tissues and deionized water.

## 2.6 Fish

At on-shore landing sites adjacent to our sampling stations, we obtained fish specimens from fishermen, which usually fish within a 20 km radius from their landing sites. We collected *Stolothrissa tanganicae*, *Limnothrissa miodon*, *Lates stappersii*, *Lates microlepis*, *Lates mariae*, and *Lates angustifrons* and processed them according to the standard protocol described in Junker et al. [35]. For stable isotope analysis, we sampled the dorsal muscle using clean scalpels and forceps and removed the skin.

Live fish samples were collected under the approved University of Wyoming IACUC protocol #20160602CW00241-01. This protocol was approved by the University of Wyoming IACUC committee with written consent and annual re-approval with written consent.

## 2.7 Isotopic and elemental analysis of solids

All solid isotope samples (POM, zooplankton, *P. spekii*, and fish) were oven-dried at ~60 ˚C for at least 24 h after collection and then packed in aluminium foil or small sample tubes. Dried samples were stored at room temperature and shipped to Switzerland. At Eawag Kastanienbaum, we fumed the POM samples for 48 h under HCl atmosphere to remove inorganic carbon. Fish and *P. spekii* samples were ground to fine powder using a Qiagen Tissuelyzer II. We measured the C and N elemental and isotopic compositions with an EA-IRMS (vario PYRO cube, Elementar coupled with an IsoPrime IRMS, GV Instruments). *Acetanilide #1* (Indiana University, CAS # 103-84-4) was used as an internal standard. The isotopic ratios of the samples are reported in the delta notation VPDB for carbon and air for nitrogen. Standard and sample reproducibility was generally better than 0.2 ‰ for $\delta^{13}$C and 0.5 ‰ for $\delta^{15}$N and highest for fish tissue (0.1 ‰ for $\delta^{13}$C and 0.2 ‰ for $\delta^{15}$N).

## 2.8 Lipid content of fish muscle tissue

For a subsample of *Stolothrissa* individuals, which exhibited the largest range in C:N ratios, we measured the total lipid content to test whether a high C:N ratio effectively translates to a higher amount of lipids in fish tissue. Total lipid content was determined gravimetrically using an extraction procedure following Chen, Shen & Sheppard [69]. In brief, ~1 mg of dried fish muscle powder was weighed into a pre-combusted glass vial, and 1 mL of 2:1 (vol:vol) dichloromethane:methanol solution was added. The sample was then ultrasonicated for 10 min. The lower phase was transferred to another pre-combusted, pre-weighed glass vial and evaporated in a heat block. The entire procedure was repeated two more times, and the resulting dry lipid mass weighed to the nearest 0.001 mg.

## 2.9 Data analysis

For calculating the depth-integrated isotopic values of POM, we normalized each sample for the phytoplankton abundance at the respective depth. We corrected the $\delta^{13}$C of non-lipid-

extracted animal tissue for its lipid content according to Post et al. [56]. We estimated the lipid content in fish tissue according to the model from the same study. For comparing the isotopic composition and the C:N ratios of fish samples between different regions, we selected individuals from sites where samples were available from both campaigns (north: landing sites adjacent to stations 1 and 2; south: landing sites adjacent to stations 7 and 9). From this subset, we additionally selected the individuals from the 50 mm size range with the highest overlap across regions and sampling campaigns for each species to minimize size-specific effects (S1 and S2 Figs; [70]). The chosen size ranges were 40–90 mm for *Stolothrissa*, 75–125 mm for *Limnothrissa*, and 200–250 mm for *Lates stappersii*. For the same analyses and reasons, we selected *P. spekii* from the exact same sites. The Bayesian ellipses in the isotopic space were calculated using the R package SIBER [71]. Statistical differences between subsets, e.g. between samples from the north and south basins, were tested with Mann-Whitney-U Tests, unless specified otherwise.

We compiled our data set in a summary table (S1 Table), which includes the physicochemical properties of the water column, the plankton densities as well as the isotopic and elemental composition of the analysed food web members. For calculating the average values and standard deviations for each basin and season, we selected subsets from the data set as described above, i.e. the resulting values represent the data shown in the main figures (Figs 3–6). Moreover, to test to what extent the physicochemical and biological variables correlate, we chose the five sites with the highest possible overlap across all variables (Sep/Oct: stations 1, 2, 6, 7, and 9; Apr/May: stations 1, 2, 4, 7, 8). The data represent either depth-integrated values or averages per site. In the resulting data set were 24 gaps compared to a total of 280 data points (10 sites and 28 variables). Gaps at the northern (station 1) or southern (station 9) extremities of the lake were filled by assuming the same value as from the neighbouring site. Other gaps were filled by calculating the average value between the two neighbouring sites (S2 Table). We produced the correlation matrixes using the R package corrplot [72] and calculated the Spearman's rank correlation coefficient (some variables were not normally distributed; Shapiro-Wilk-Test, $p < 0.05$). Moreover, we performed principal component analyses (PCAs) on the C and N isotopic compositions as well as the C:N ratios of all food web members. The PCAs were calculated using the "prcomp" command in R. The data were zero centered and scaled to have unit variance. For the PCA on the full data set, we log transformed the variables due to the large variation between plankton community and bivalve or fish tissue samples.

## 3. Results

### 3.1 Hydrodynamic and biogeochemical conditions in Lake Tanganyika

Lake Tanganyika showed clear north-south differences in both hydrodynamics and biogeochemistry; the north-south gradients were stronger at the end of the dry season (September/October 2017) compared to the end of the rainy season (April/May 2018).

In Sep/Oct, both higher Sc and $N^2$ values point to a higher level of water column stratification in the north compared to the south basin (Fig 3a and S1 Table). Most prominently, the thermocline was heavily uplifted towards the south and completely absent at station 9 at the southern end of the lake (Fig 3b), indicating the deep vertical mixing from the earlier dry season. In Apr/May, significantly higher Sc and $N^2$ values, a pronounced thermocline throughout the lake, as well as lower surface temperatures (Δ~0.5 ˚C) indicate that the lake was overall more stratified during that period, with a lesser degree of upwelling/mixing in the south (Fig 3a–3d; $p < 0.05$).

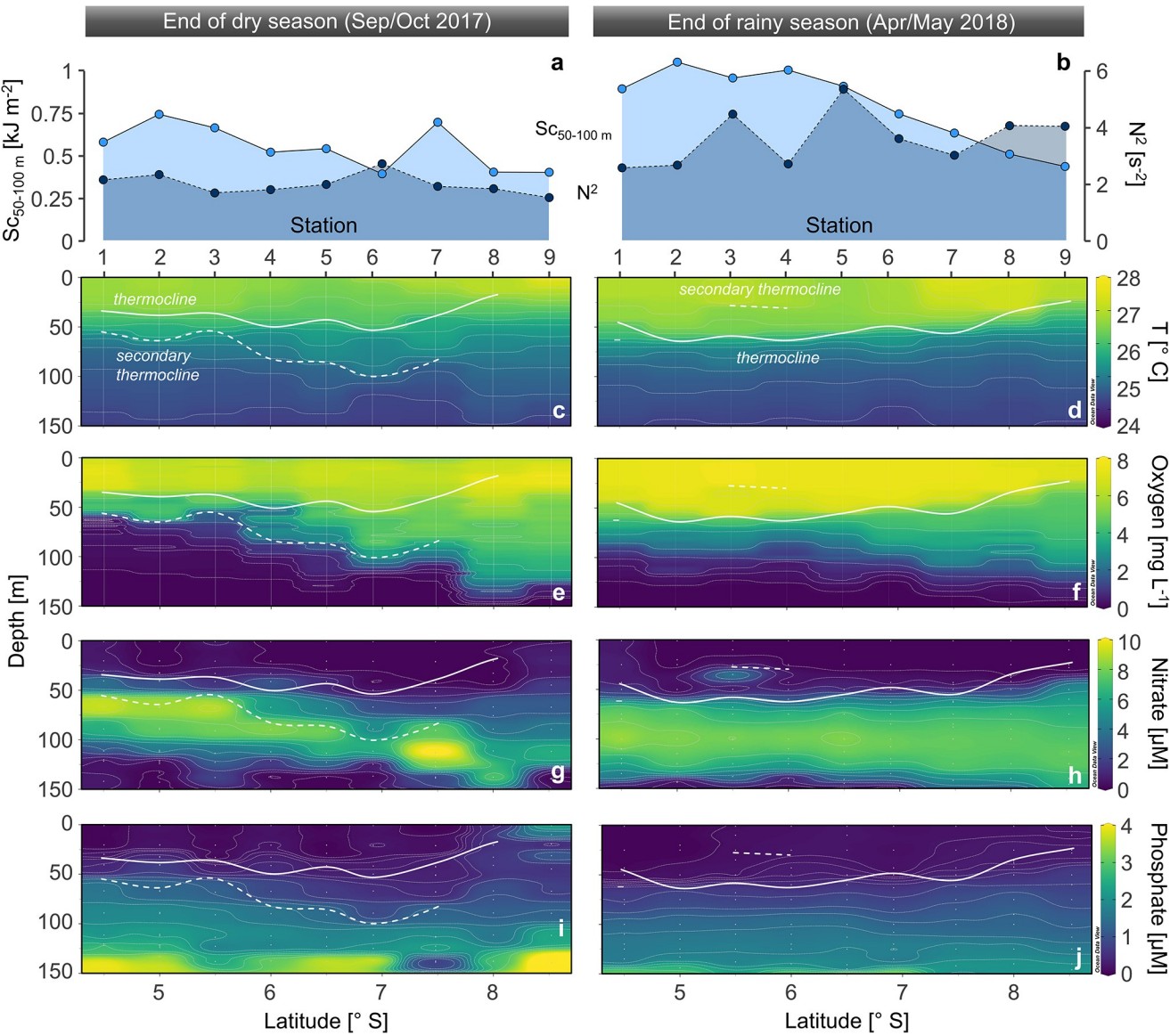

**Fig 3. Physical and chemical properties of Lake Tanganyika along our north-south transects (from station 1–9) at the end of the dry season (left) and the end of the rainy season (right).** (a,b) Schmidt stability (Sc) of the 50–100 m depth interval and buoyancy frequency of the primary thermocline ($N^2$). Distribution of (**c,d**) temperature (T), (**e,f**) dissolved oxygen, (**g,h**) nitrate, and (**i,j**) phosphate. The solid white line depicts the thermocline, whereas the dashed white line represents less pronounced secondary thermoclines. No clear thermocline had formed at station 9 at the end of the dry season. Samples are indicated by vertical lines (continuous profiles) or points (discrete samples).

Similarly the distributions of both dissolved oxygen and nutrients exhibited starker north-south contrasts in Sep/Oct compared to Apr/May. For instance, the oxyline deepened from ~60 m at station 1 down to ~140 m station 9 in Sep/Oct, whereas the oxycline location only varied from ~80–120 m in Apr/May (Fig 3e and 3f). In the surface waters, the concentrations of both nitrate and phosphate reached their overall maximum of 0.7 µM (nitrate) and 2.3 µM (phosphate) at the southern end of the lake in Sep/Oct (Fig 3g–3i). Ammonium concentrations were below limit of detection in the upper 100 m during both sampling campaigns (S3 Fig).

## 3.2 Particulate matter, phytoplankton, and zooplankton

In line with the hydrodynamics and biogeochemistry, most of our measured plankton parameters also exhibited systematic latitudinal and seasonal patterns, with generally more pronounced north-south differences in Sep/Oct compared to Apr/May. For instance, the concentration of chlorophyll-*a* was significantly higher in the south compared to the north basin during Sep/Oct (~51 mg m$^{-2}$), whereas it was at a similarly low level of ~42 mg m$^{-2}$ in both basins during Apr/May (Fig 4a). By contrast, the abundance of medium- to large-celled phytoplankton (>10 μm) was lower in the south compared to the north basin during both campaigns (Fig 4b). $\delta^{13}$C-POM was higher in the south compared to the north basin during both campaigns, with none of the differences being significant (Fig 4c). Opposite to all other plankton parameters, $\delta^{15}$N-POM showed similar values among the north and south basins in Sep/Oct (1.6 and 1.4 ‰, respectively) and strong differences in Apr/May, with a minimum in the north (-1.0 ‰) and higher values in the south (1.0 ‰, Figs 4d, 5a and 5b and S2 Table).

Compared to the analyzed phytoplankton variables, zooplankton parameters showed more consistent patterns. For example, zooplankton from both the 95 μm and 250 μm size fractions

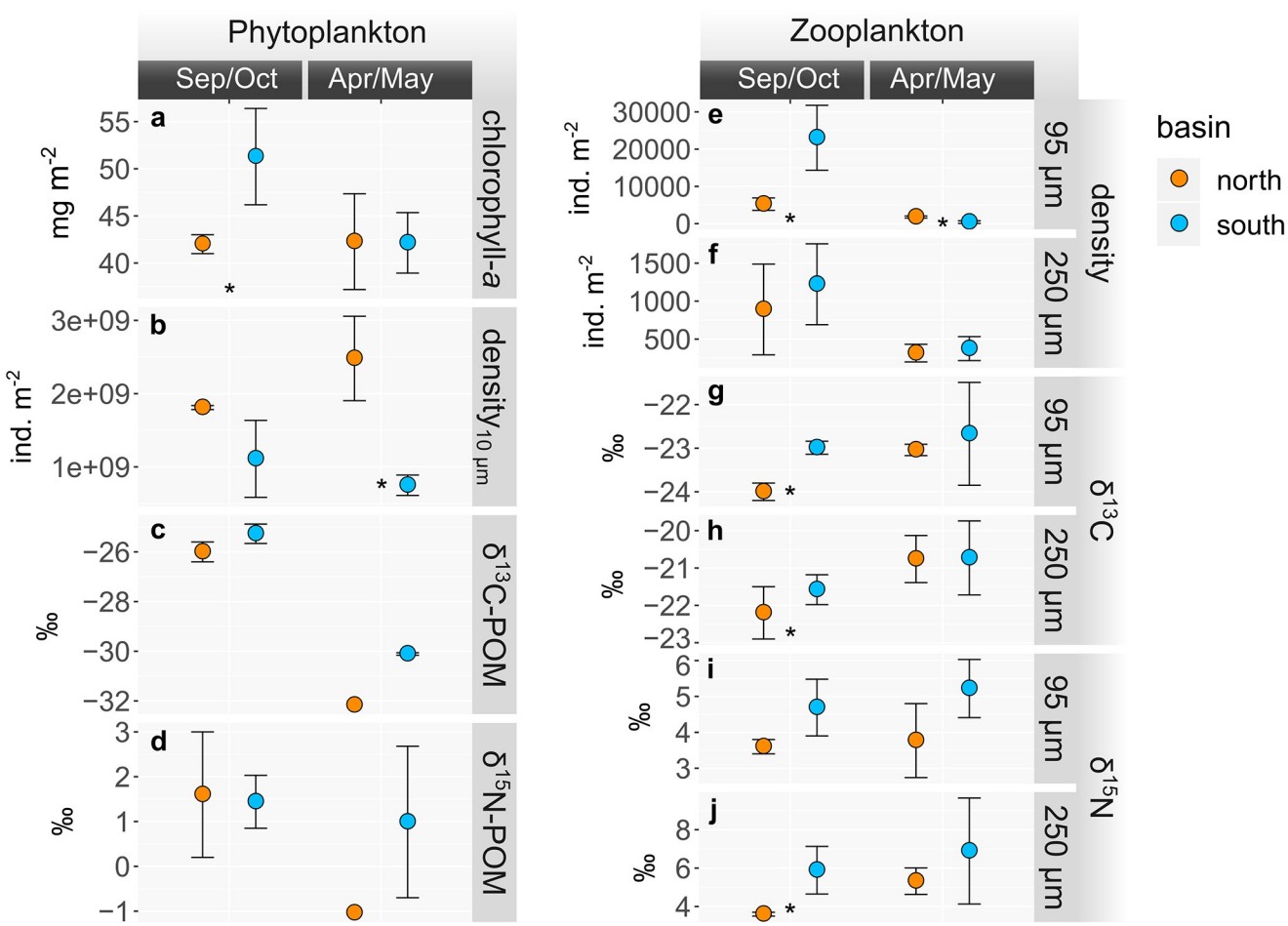

**Fig 4. Phyto- (left) and zooplankton (right) parameters sampled in the north and south basins of Lake Tanganyika at the end of the dry season (Sep/Oct 2017) and the end of the rainy season (Apr/May 2018).** (**a**) Depth-integrated chlorophyll-a concentration, (**b**) phytoplankton (>10 μm) abundance, (**c**) depth-integrated $\delta^{13}$C, and (**d**) $\delta^{15}$N values of POM. (**e,f**) Abundances, (**g,h**) $\delta^{13}$C, and (**I,j**) $\delta^{15}$N values of the 95 μm and 250 μm zooplankton fractions, respectively. Values represent averages with standard deviations. Sample sizes are given in S1 Table (n ≤ 3). * *p*-value ≤ 0.1 between north and south.

was more abundant in the south compared to the north basin in Sep/Oct, whereas the abundances were more similar among the basins in Apr/May (Fig 4e and 4f). In accordance, the $\delta^{13}$C values of both size fractions were significantly higher in the south compared to the north basin in Sep/Oct, and similar between north and south in Apr/May (Fig 4g and 4h and S1 Table). By contrast, the $\delta^{15}$N values of both size fractions were higher in the south than in the north during both seasons, whereby the difference was significant in Sep/Oct (Figs 4i, 4j, 5c and 5d).

### 3.3 Isotopic composition of bivalve and fish

In line with the analyses of the hydrodynamics, biogeochemistry, and planktonic food web, the $\delta^{13}$C of *P. spekii* and all fish species was highest in the south during Sep/Oct and at a similarly low level otherwise. For instance, northern and southern samples of *P. spekii* diverged significantly in Sep/Oct ($p < 0.01$), whereas the $\delta^{13}$C values from the two basins overlapped in Apr/May with northern samples nesting fully within the range of southern samples (Fig 5e and 5f).

Likewise, the mean $\delta^{13}$C values of the largely planktivorous sardines *Stolothrissa* and *Limnothrissa* as well as the zooplanktivorous and piscivorous *Lates stappersii* diverged significantly by approximately 0.7 ‰ between the north and the south in Sep/Oct ($p < 0.001$, Fig 5g, 5i and 5k). By contrast, the differences in mean values were completely erased or slightly reversed in Apr/May (Fig 5h, 5j and 5l). Moreover, we found the highest $\delta^{13}$C values in the south for all three species, with averages of -20.7 ‰, -20.3 ‰, and -20.4 ‰ for *Stolothrissa*, *Limnothrissa*, and *Lates stappersii*, respectively.

In contrast to *P. spekii* and *Stolothrissa*, the southern samples of *Limnothrissa* and *Lates stappersii* showed 0.2 and 0.4 ‰ lower $\delta^{13}$C values compared to the northern one in Apr/May, respectively. This difference was not significant for *Limnothrissa* ($p > 0.05$) and was significant for *Lates stappersii* ($p < 0.001$). It is worth noting that the subsets of *Limnothrissa* and *Lates stappersii* from Apr/May included in our analysis were slightly unbalanced with respect to size, i.e. with samples from the north being larger compared to samples from the south (S2 Fig). Since larger individuals in these two species tend to have less depleted $\delta^{13}$C values (S4 Fig), the $\delta^{13}$C values of the northern samples, and thus the difference between the basins, are slightly overestimated here. Samples from the central basin were nested within the distributions from the north and south for the two sardines and *L. stappersii* (S5 Fig).

We had fewer samples of the large predators *Lates microlepis*, *Lates mariae*, and *Lates angustifrons*, preventing an in-depth statistical analysis, but the results hint at similar patterns. Across our entire data set, these three species showed the highest $\delta^{13}$C values, with most observations being heavier than -21 ‰ (S4g, S4i, S4k and S6 Figs). In Sep/Oct, the $\delta^{13}$C values of both *Lates microlepis* and *Lates angustifrons* specimens from the south were >0.5 ‰ higher than the individuals from the northern specimens. In Apr/May, samples from both the north and south basins were only available for *Lates mariae*. Here, the $\delta^{13}$C values from both basins varied within the same range and their averages differed only slightly (north: -20.1 ‰; south: -20.3 ‰).

Similar to zooplankton and POM, *P. spekii* showed consistently higher $\delta^{15}$N values in the southern basin, reaching 1.7 and 2.1 ‰ on average in Sep/Oct and Apr/May, respectively, compared to the northern basin with averages of 1.5 and 1.3 ‰, for the two campaigns (Fig 5e and 5f). In contrast to $\delta^{13}$C, we observed no systematic seasonal or regional differences in fish $\delta^{15}$N values (Fig 5g–5l and S6 Fig). *Stolothrissa* exhibited the lowest values with basin-wide averages spanning from 4.9 to 5.1 ‰. The other sardine species, *Limnothrissa*, had markedly higher values (means: 5.4–5.6 ‰). The variation appeared to be neither related to site nor season, and only to a small extent to size for both sardine species (S4b and S4d Fig). By contrast,

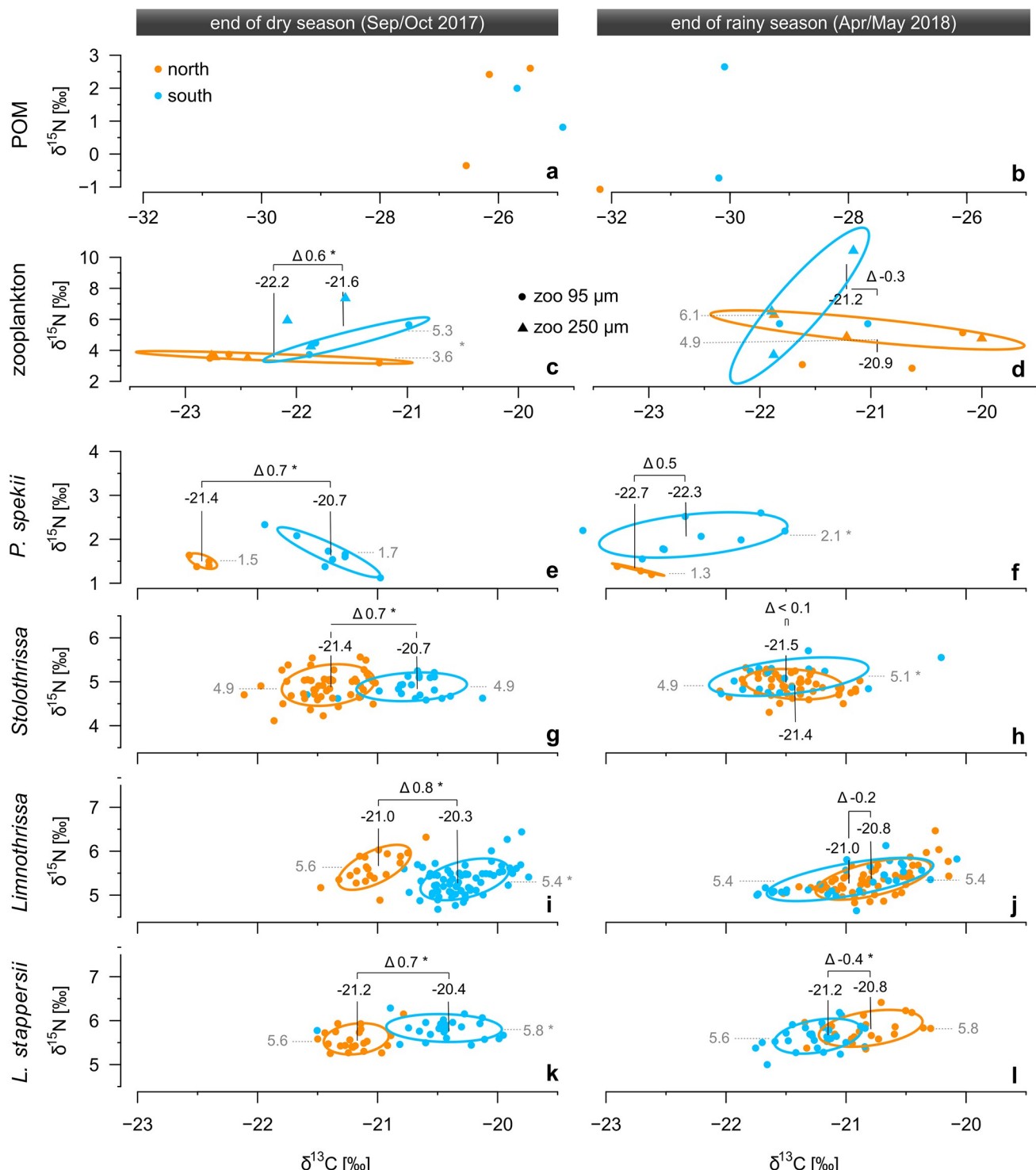

**Fig 5. Carbon (normalized for C:N mass ratio according Post et al. [56]) and nitrogen stable isotope signatures of the major pelagic food web members.** (**a,b**) POM (**c,d**) zooplankton, (**e,f**) the bivalve *Pleiodon spekii* as well as the fish (**g,h**) *Stolothrissa tanganicae*, (**i,j**) *Limnothrissa miodon*, and (**k,l**) *Lates stappersii* at the end of the dry season (left) and the end of the rainy season (right). Orange dots represent the northern basin and blue dots represent the southern basin. Numbers indicate the mean $\delta^{13}C$ (black) and $\delta^{15}N$ (grey) values from each basin and $\Delta$ denotes the $\delta^{13}C$ difference between the southern and northern mean values. Ellipses encompass approximately 67% of the data for plankton samples (a-d) and 95% of the data from each basin for tissue samples (e-l). Note different axis limits. * $p$-value $< 0.05$ between north and south.

the δ[15]N values of the larger *Lates* species were primarily related to size (S4f, S4h, S4j and S4l Fig). None of the *Lates* species revealed clear basin-wide differences in δ[15]N values between the basins or seasons (Fig 5k and 5l and S6 Fig).

## 3.4 C:N ratios and estimated lipid content

There were no clear differences between north and south in any of the organisms (Fig 6). By contrast, we observed strong changes in C:N ratios between Sep/Oct and Apr/May at lower and middle trophic levels, but not at high trophic levels. *Pleiodon spekii* and zooplankton showed consistently lower C:N ratios in Sep/Oct compared to Apr/May (Fig 6a, 6c and 6e).

Contrary to the trends in *P. spekii* and zooplankton, the sampled fish species showed a tendency towards higher C:N ratios in Sep/Oct (Fig 6b, 6d and 6f). This trend vanished with increasing trophic level. The C:N ratios of the planktivorous clupeid *Stolothrissa* decreased significantly ($p < 0.001$) between Sep/Oct and Apr/May in both the northern and southern basins (medians of 3.65 versus 3.24 and 3.49 versus 3.19, respectively). *Limnothrissa* revealed significant changes in the north (Sep/Oct median: 3.23; Apr/May median: 3.18; $p < 0.001$), whereas the pattern was reversed and insignificant in the south (Sep/Oct median: 3.19; Apr/May median: 3.21; p > 0.05). By contrast, the C:N ratios of *Lates stappersii* varied only within a narrow range, with medians spanning from 3.18 to 3.21 and no significant differences between seasons.

Using the model from Post et al. [56], we estimated the lipid contents in the dorsal muscle tissue of the investigated fish species from their C:N ratios (Fig 6b, 6d and 6f). This analysis

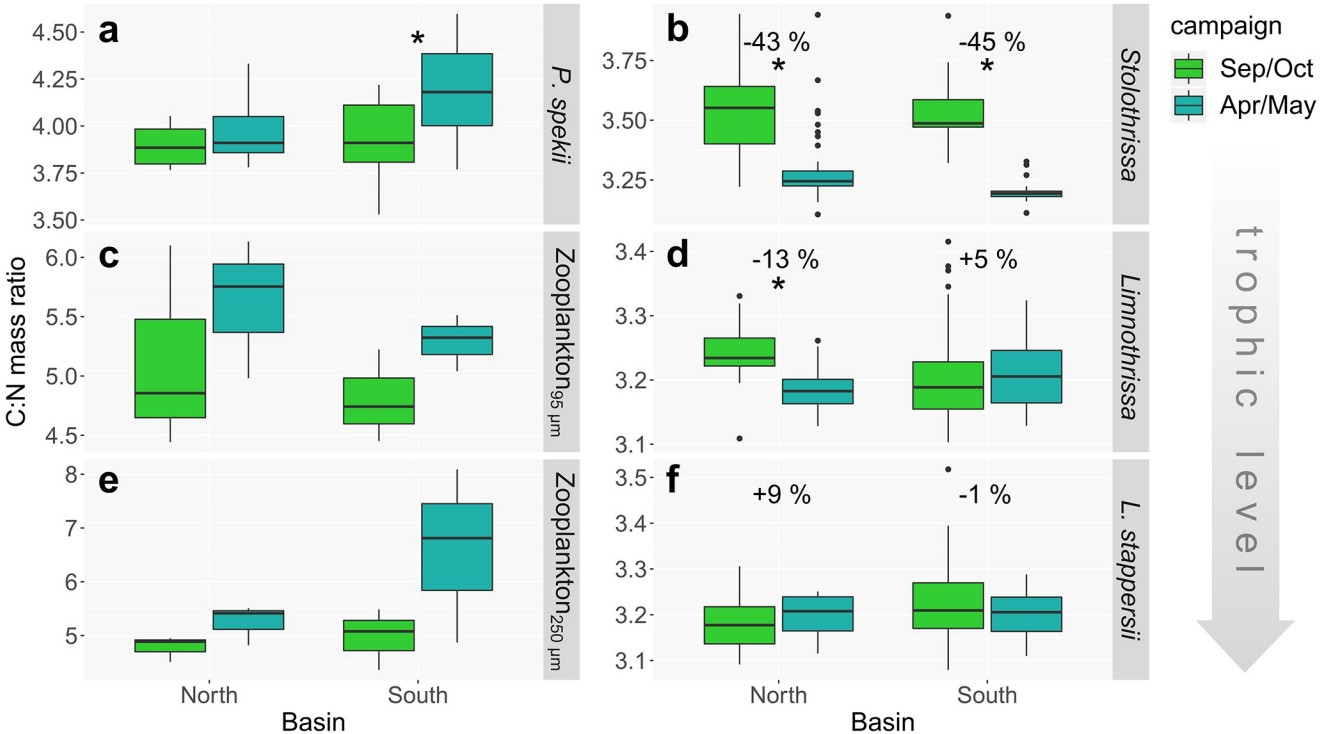

**Fig 6. Mass C:N ratios of primary consumers (left) and fish tissue (right) for the different sampling campaigns and basins of Lake Tanganyika.** (**a**) *Pleiodon spekii*, (**c, e**) zooplankton, (**b**) *Stolothrissa tanganicae*, (**d**) *Limnothrissa miodon*, and (**f**) *Lates stappersii*. Numbers depict the % change in estimated lipid content according to Post et al. [56]. Note varying y-axis scaling. * *p*-value < 0.05 between Sep/Oct and Apr/May.

suggests that the lipid content of *Stolothrissa* almost halved from 5.2% in Sep/Oct to 3.0% Apr/May in the north (Δ = -43%) and from 4.7 to 2.6% in the south (Δ = -45%), respectively. *Limnothrissa* exhibited a decrease from 2.9 to 2.5% (north; Δ = -13%) and an increase from 2.5 to 2.7% (south; Δ = +5%). *Lates stappersii* showed the lowest differences with an increase from 2.5 to 2.7% in the north (Δ = +9%) and similar values of 2.7% in the south (Δ = -1%). A gravimetric determination of lipid content from selected *Stolothrissa* samples confirmed that higher C:N ratios translate into higher lipid contents (linear regression, $R^2$ = 0.91, $p < 0.01$, $n$ = 5; S7 Fig).

## 3.5 Correlations across the data sets

To test the statistical significance of the observed north-south patterns, we calculated two correlations matrixes across the data sets for both Sep/Oct (Fig 7a and S8 Fig) and Apr/May (Fig 7b and S9 Fig). In line with the congruent patterns, the $\delta^{13}C$ values of all food web members correlated positively in Sep/Oct. Most relationships were not significant, except *Stolothrissa* and *P. spekii*, *Limnothrissa* and zooplankton (95 μm), as well as *Lates stappersii* and *Stolothrissa*. In Apr/May, by contrast, the food web members did not correlate systematically and none of the relationships were significant.

In Sep/Oct, the $\delta^{13}C$ values also revealed systematic relationships with plankton and physical variables. The $\delta^{13}C$ values correlated positively with chlorophyll-*a* as well as the zooplankton abundances of the 25 μm and 95 μm fractions. Of those, the relationships between $\delta^{13}C$-POM and chlorophyll-*a* as well as between the $\delta^{13}C$ of *P. spekii* or *Stolothrissa* and the zooplankton abundance (95 μm) were significant. By contrast, the $\delta^{13}C$ values correlated

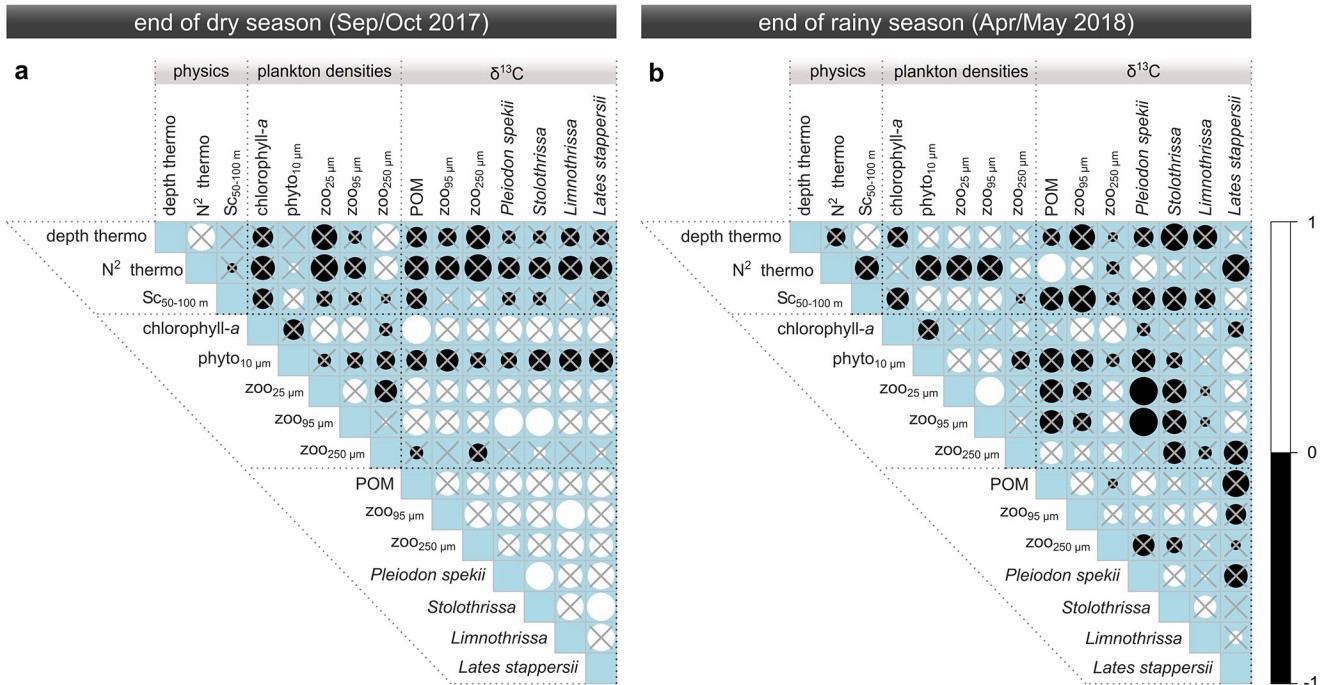

**Fig 7. Spearman-rank correlation matrix of physical, plankton, and $\delta^{13}C$ variables for (a) the end of the dry season and (b) the end of the rainy season.** For each season, we selected the five stations across the north-south transect with the highest overlap among all variables (Sep/Oct: stations 1, 2, 6, 7, 9; Apr/May: stations 1, 2, 4, 7, 8; S2 Table). Insignificant correlations ($p > 0.05$) are marked by grey crosses. Depth thermo: depth of the primary thermocline; $N^2$ thermo: buoyancy frequency of the primary thermocline; $Sc_{50-100\,m}$: Schmidt stability of the 50–100 m depth interval; $Phyto_{10\,\mu m}$: phytoplankton abundance of the >10 μm size fraction; $Zoo_{25/95/250\,\mu m}$: zooplankton parameters of the >25, >95, or >250 μm size fractions.

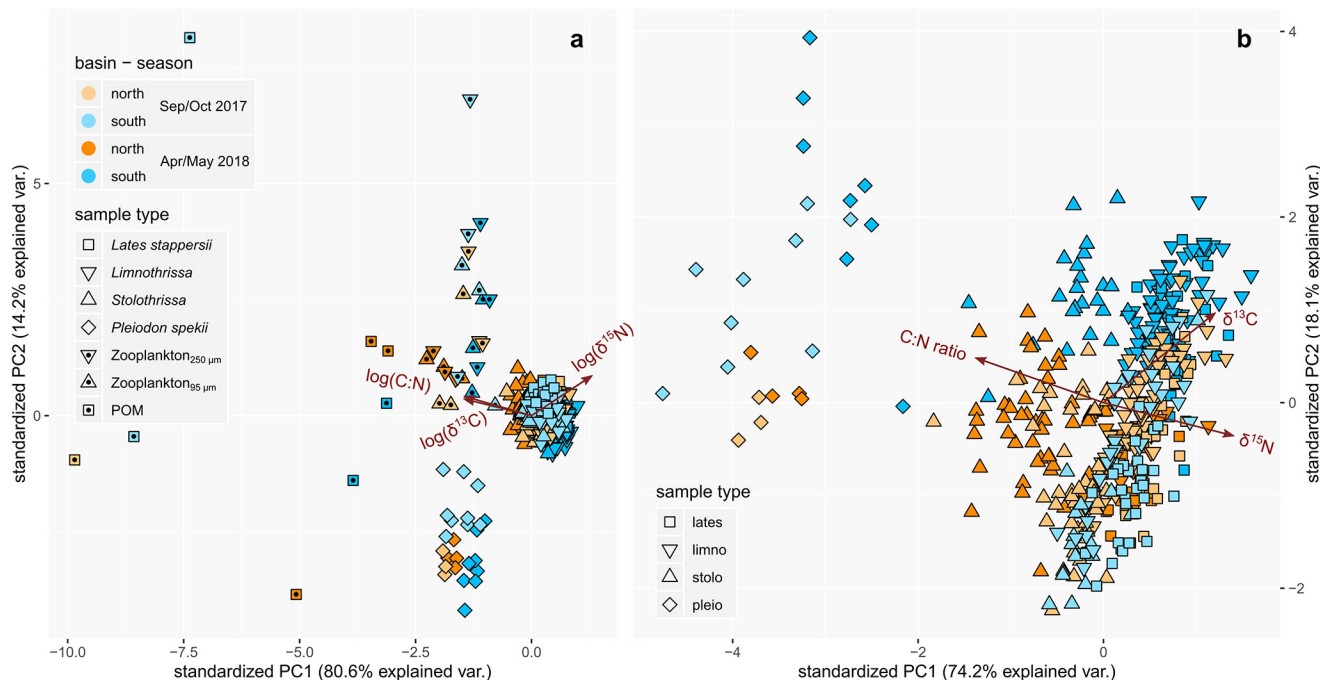

**Fig 8. Principal component analyses of the C and N isotopic compositions as well as mass C:N ratios of the food web.** Analyses were done (**a**) on all food web members and (**b**) on bivalve and fish tissue samples. Variables in panel a are log transformed.

negatively with the depth and $N^2$ of the thermocline as well as with the phytoplankton (>10 μm) densities, with none of the relationships being significant. In Apr/May, none of the other variables correlated in a congruent fashion with the $\delta^{13}$C the food web. There were no systematic correlations across all food web members for $\delta^{15}$N or C:N ratios in either of the seasons (S8 and S9 Figs). A PCA analysis confirmed that, within each food web member, the north-south differences during Sep/Oct were primarily determined by $\delta^{13}$C (Fig 8).

Lastly, the total zoo- and phytoplankton abundances were negatively correlated in Sep/Oct, i.e. phytoplankton (>10 μm) decreased, when zooplankton increased ($rho$ = -0.58, $p \sim 0.1$, Spearman's rank correlation; Fig 7a). No significant correlation between phyto- (>10 μm) and zooplankton was found in Apr/May ($rho$ = 0.38, $p > 0.3$, Spearman's rank correlation; Fig 7b).

## 4. Discussion

Our results show that upwelling and mixing in the south basin of Lake Tanganyika stimulate biological productivity. Upwelling and mixing furthermore coincide with higher $\delta^{13}$C values of phytoplankton, potentially due to the accelerated primary productivity rates. Our data show that these regional (i.e. basin-scale) differences in the $\delta^{13}$C of primary producers propagate through the food web to higher trophic levels. On the background of recent genetic studies, we discuss to what extent the resulting isotopic differences between north and south may be used to assess the connectivity and ecology of the pelagic fish stocks.

### 4.1 Effect of upwelling and mixing on the isotopic composition of the planktonic food web

Upwelling and convective mixing moderate the transport of nutrients to the surface waters, and thus drive biological productivity during the dry season in Lake Tanganyika. In this study,

we compare two contrasting hydrodynamic situations: First, the period of re-establishing water column stratification at the end of the dry season (Sep/Oct). During this time, stratification was weaker and the thermocline was still absent at the southernmost station, enabling particularly high nutrient fluxes in the upwelling-driven south basin. Second, the period of lake-wide stratification at the rainy season-dry season transition (Apr/May). Here, the water column experienced stronger stratification from the preceding rainy season and beginning trade winds initiated the upwelling in the south, resulting in overall lower nutrient fluxes with a maximum in the south.

In line with the hydrodynamic regime, our data suggest that the planktonic food web was most productive during the seasonal upwelling/mixing in the south basin. For example, the concentrations of chlorophyll-*a*, a proxy for photosynthetic activity, reached their overall maximum in the south at the end of the dry season. By contrast, we detected a southward decrease in the abundance of large-sized phytoplankton (>10 μm)–a pattern that has previously been observed in Lake Tanganyika [73]–pointing to additional ecological controls, such as zooplankton grazing or competition within the phytoplankton community. For instance, a cyanobacterial bloom may explain the high phytoplankton (>10 μm) abundances in the north in Apr/May [58]. Moreover, the nano- and pico size fractions (<10 μm) are more competitive under nutrient-rich conditions and therefore dominate the phytoplankton community in south of Lake Tanganyika [74, 75]. As a result of their high densities, total phytoplankton abundance and biomass is generally highest during the dry season upwelling in the south [21, 24, 76].

In addition, upwelling/mixing and the subsequent stimulation of primary productivity, place important bottom-up control on the abundances of zooplankton in Lake Tanganyika. This was evidenced by high zooplankton abundances in the dry season with maxima in the southern basin (Fig 4; [28, 77]), which in turn sustain the growth of the pelagic fish populations [4, 25, 26, 78]. The high zooplankton abundances can in turn exert top-down control over phytoplankton, which is indicated by the negative correlation between the phytoplankton (>10 μm) and zooplankton abundances in Sep/Oct. This grazing effect may have also been responsible for the absence of stronger differences in chlorophyll-*a* between Sep/Oct and Apr/May. Zooplankton abundance, and thus grazing pressure, exhibited no clear latitudinal trend during the Apr/May campaign, when the lake-wide stratification was stronger, and the north-south gradients in nutrient availability and biological productivity were not as pronounced.

The varying hydrodynamics were also associated with distinct $\delta^{13}$C-POM signatures that may reflect differences in primary productivity. The average $\delta^{13}$C was ~5 ‰ heavier in Sep/Oct compared to Apr/May, and the $\delta^{13}$C increased by ~2 ‰ from north to south during both campaigns (Fig 4c). In Lake Tanganyika, previous studies also revealed heavier $\delta^{13}$C-POM values in the dry season [53–55], even though the differences were smaller (max. 3.1 ‰) than in our study (max. 6.7 ‰), possibly due to the varying timing of the sampling. These differences in $\delta^{13}$C-POM likely reflect the well-documented changes in primary production [22, 76, 79], where heavier isotopic signatures in POM mirror the incorporation of a larger $^{13}$C fraction by higher photosynthesis and cell growth rates [46, 47] and a stronger drawdown of the DIC pool [48–50, 80]. The links between stratification, vertical nutrient supply, and primary productivity in Lake Tanganyika are well established [14–16, 22] and several studies have used the $\delta^{13}$C of sediment POM to infer primary productivity [11, 16]. Correspondingly, the $\delta^{13}$C-POM positively correlated with chlorophyll-*a* concentrations ($p < 0.05$, Fig 4). In further support, our own $CO_2$ fixation rate measurements, done during Apr/May, show evidence for higher productivity rates in the southern basin at station 7 in the south compared to station 2 in the north (S10 Fig). On the other hand, upwelling of intermediate waters will not only supply

nutrients, but also isotopically light DIC (depleted by ~1 ‰; S11 Fig; [67, 81]). Although this mechanism will slightly dilute $^{13}C$ enrichment, our proposed mechanism of higher primary productivity is apparently strong enough to overcome this depletion in $\delta^{13}C$-POM, ultimately leading to higher $\delta^{13}C$-POM values when upwelling/mixing is stronger.

The analogous pattern in $\delta^{15}N$-POM implies that upwelling and mixing may have influenced the N sources of primary producers, where lighter values are typically interpreted as inputs from N fixation [80, 82]. POM $\delta^{15}N$ values increased from -1.0 ‰ at station 3 to 2.7 ‰ at station 9 in the south in Apr/May, concurrent with a decrease in filamentous, N-fixing cyanobacteria [58], whereas it fluctuated with slightly higher values (-0.3–2.6 ‰) in Sep/Oct devoid of a latitudinal or phytoplankton composition related pattern (S2 Table). We also observed no correlation between the presence of surface nitrate and $\delta^{15}N$-POM, which may have induced fractionation effects during nitrate-uptake. When free nitrate remains, the phytoplankton community does not represent a complete sink of the upward diffusing nitrate, i.e. the residual nitrate should be isotopically heavy and phytoplankton relatively light. In line with the higher density of N-fixing cyanobacteria [24, 58, 75], the generally lighter $\delta^{15}N$-POM in Apr/May ($\Delta$-1.4 ‰) point at higher inputs from N fixation compared to Sep/Oct, when nutrient fluxes are higher due to upwelling/mixing.

The isotopic composition of the zooplankton community also followed the north-south patterns, but some extreme values point at additional influencing factors than the signal from the base of the food web, i.e. POM. The $\delta^{13}C$ values in our zooplankton samples oscillated between -24.7 and -21.1 ‰, in accordance with previous isotope surveys. The $\delta^{15}N$ values from the north were also in agreement with other studies, whereas the maxima in the south, where previously no isotopic characterization of the food web was undertaken, exceeded earlier reports by min. 2.7 ‰ (Fig 4; [53–55, 70]). The high intra-basin variability in $\delta^{13}C$ and $\delta^{15}N$ as well as the high absolute values relative to other members of the food web, with some zooplankton $\delta^{15}N$ exceeding top predator fish $\delta^{15}N$ values, may be in part attributable to varying zooplankton community compositions [54, 70]. Such values from a pooled zooplankton sample are not unexpected, because zooplankton communities consist usually of members from several trophic levels [46, 54, 77, 78, 83, 84], and our samples represent batch samples from entire zooplankton communities formed by many different species, genera, and families. In addition, the zooplankton community is notoriously hard to sample and standard netting techniques do not capture fast swimmers such as shrimps efficiently, therefore often underestimating their abundances [28]. Indeed, shrimps only made up minor proportions in our samples (S12 Fig) and previous work showed that they have $\delta^{13}C$ and $\delta^{15}N$ values lower than our community isotope values [54, 55, 70]. However, reported $\delta^{15}N$ values of individual zooplankton taxa, including detrivorous jellyfish and fish larvae, do not exceed 5.9 ‰ [54, 55, 70] and therefore fail at explaining the high $\delta^{15}N$ in our measured community isotope samples from the south (>10 ‰; S2 Table). In line with our results, earlier reports of bulk community samples found high $\delta^{15}N$ values between 6 and 8 ‰ [53], raising questions about the utility of using bulk community samples. Combining the taxonomic assessment of the community with an isotopic characterization of individual zooplankton taxa would thus be valuable in future food web studies.

In summary, our results point to a pivotal role of nutrient upwelling and mixing for sustaining the high biological productivity in the south basin during the dry season. Upwelling-related increases in primary productivity and decreases in N fixation likely resulted in markedly heavier planktonic $\delta^{13}C$ and slightly heavier $\delta^{15}N$ values in the south (Fig 9a). The consistently higher zooplankton $\delta^{13}C$ and $\delta^{15}N$ values in the southern basin in Sep/Oct may reflect the isotopic imprint of the upwelling/mixing, but clearer trends may be masked to some extent by concomitantly shifting community composition effects.

**Fig 9. Schematic synthesizing the main conclusions and hypotheses of the study.** (**a**) Biological productivity of phyto- and zooplankton based on abundance and $\delta^{13}$C data, which were used to (**b**) infer the distribution of regional fish stocks from their $\delta^{13}$C signatures. (**c**) The regional isolation of the fish stocks at seasonal timescales does not translate into suppressed gene flow at generational timescales, as indicated by a lack of regional genetic structure in these species. (**d**) The regional fish stocks as well as different genetic clusters did not exhibit systematic differences in $\delta^{15}$N. (**e**) The clupeid *Stolothrissa* exhibited strong seasonal changes in C:N, i.e. lipid content, indicating lipid storage after the productive dry season. *results from Junker et al. [35] and Rick et al. [36].

## 4.2 Isotopic imprints from upwelling and mixing reveal regional fish stocks

The regionally and seasonally varying hydrodynamic conditions also determine the C isotopic compositions of organisms higher in the food web, through the incorporation of phyto- and zooplankton prey. For instance, we used the filter-feeding bivalve *P. spekii* as a reference organism for the seasonal phytoplankton isotopic signals. Tissue turnover in bivalves is significantly slower than in phyto- and zooplankton [53, 85] and similar to the muscle half-life in fish, which ranges from a few weeks in juveniles to several months in adults [86–88]. Therefore, *P. spekii* integrates the isotopic signals from its food over a longer time span; it lives for at least five years at a stationary location and was successfully used to record upwelling events in Lake Tanganyika [89]. At the end of the dry season, we find that the *P. spekii* samples from the upwelling-driven south are significantly enriched in $^{13}$C compared to the northern samples (Δ1.1 ‰). Lake-wide stratification over the rainy season on the other hand resulted in converging δ$^{13}$C values between samples from both basins in Apr/May (Fig 5e and 5f). The uniform *P. spekii* δ$^{13}$C values in Apr/May indicate that the north-south differences, which we observed for POM and zooplankton during this time, were likely a result of the commencing upwelling during our sampling campaign rather than a consistent seasonal value.

Seasonal cycles in δ$^{13}$C signals are common in aquatic food webs and propagate up the trophic chain [52, 90]. In our study, the δ$^{13}$C of POM, zooplankton, and *P. spekii* positively correlated with the δ$^{13}$C of all fish species in Sep/Oct, with a significant latitudinal difference of ~0.7 ‰ heavier fish tissue δ$^{13}$C in the south (Figs 5 and 7). By contrast, the aligning δ$^{13}$C values between northern and southern fish samples during the rainy season are again in agreement with the more uniform primary productivity patterns. In line with trophic dynamics, the significant correlations are direct predator-prey relationships, i.e. between *Stolothrissa* and *P. spekii* (reference for phytoplankton), *Limnothrissa* and zooplankton (95 μm), as well as *Lates stappersii* and *Stolothrissa* (Fig 7a). Despite the congruent patterns across the food web, it is not surprising that most correlations were statistically insignificant, given our sample size of five and the chosen Spearman's rank method. Rank-based tests sacrifice explanatory power in favor of not assuming normal distribution. Overall, our results point to fish stocks confined to regional foraging grounds in the respective basins, which therefore record the latitudinal isotope gradients (Fig 9b).

In light of recent genetic studies, the isotopically distinct fish stocks can only be regarded as regional on rather short seasonal time scales, though (Fig 9c). Previous high resolution population genetic work did not find evidence for genetic differentiation between the north and south basins in any of the six fish taxa investigated in this study. Instead, specimens from the north and south basins are closely related [34–36]. The limited genetic differentiation in these species is not spatially restricted, with the exception of a case in *Lates mariae*. Rick et al. [36] found one *Lates mariae* cluster confined to the extreme south end of the lake with strong genetic differentiation from individuals elsewhere in the south basin or in the rest of the lake. Thus, the genetic structure of the fish populations cannot be explained by the basin-scale dynamics. This implies that the degree of geographical isolation between north and south basin itself is insufficient to suppress lake-wide gene flow in these pelagic fish species; i.e. the fish move randomly across the lake on long term generational time scales. At least on a seasonal time scale, however, the fish reside in a region, even though the exact time scale of restricted movement is unclear.

A greater understanding of how these fishes move throughout their life cycle would further help to reconcile these patterns. Our findings are consistent with any life history involving substantial movement in the early life followed by restricted movement at later life stages (e.g. when our sampling occurred, as we did not sample juvenile fishes).

## 4.3 Does the lake-wide gene flow inhibit ecological differentiation between regional fish stocks?

Despite the absence of pronounced spatial genetic structure in either of the sardine [34, 35] or *Lates* species [36] phenotypic traits, such as diets and lipid contents, may vary between regional fish stocks in response to regionally different environments, which include a northern region with a more stable and clear water column and a plankton-rich upwelling region in the south [19, 21, 78].

However, the relatively constant average $\delta^{15}N$ of the studied fish ($\Delta < 0.2$ ‰) and *P. spekii* ($\Delta < 0.8$ ‰) among the sampling campaigns and basins indicates no strong differences in trophic level between the regional stocks. Precisely quantifying the trophic position of the fish species was difficult: first, the $\delta^{15}N$ of potential prey organisms–the bulk zooplankton community–varied by up to 6.7 ‰ within the basins and maximum values were similar to the highest fish $\delta^{15}N$ values, which raises doubt about the usefulness of comparing bulk community with tissue samples [91]; second, the $\delta^{15}N$ values of different zooplankton taxa can vary greatly, with reported values ranging from 0.1 to 5.9 ‰ in Lake Tanganyika [54, 55, 70]; and third, the exact trophic discrimination factors are not known for Lake Tanganyika [92], but appear to largely deviate from the norm [54, 70]. Together these three factors make it difficult to assess subtle dietary differences with our bulk isotope approach. In future studies, compound specific isotope analyses of amino acids may help to further constrain the trophic relationships in Lake Tanganyika's pelagic food web [93].

Nonetheless, the relatively consistent basin-wide average $\delta^{15}N$ values demonstrate that the isotopic composition of the fish's N sources does not vary substantially throughout the year and among basins, although POM and zooplankton showed a tendency towards higher values in the south. Compared to the basin-scale mixing regime and associated plankton dynamics, other factors seem to have a greater influence on the trophic levels, or $\delta^{15}N$ values, of the studied fish taxa. Accordingly, the differences in $\delta^{15}N$ between fish specimens of a similar size, that were found at the same location and time, were an order of magnitude greater (>2 ‰) than between the regions/basins (<0.2 ‰). We found no clear evidence, however, that differences in $\delta^{15}N$ among fish individuals were linked to the genetic clusters of *Limnothrissa* or the four *Lates* species that genetic work previously detected (Fig 9d and S13 Fig).

Fish use lipids to store energy during times of abundant food supply to bridge resource limited periods [38, 39]. In congruence with the absence of basin-scale genetic structure and differences in $\delta^{15}N$, we found no regional differences in C:N ratios as proxy for lipid content (S7 Fig; [56, 57, 94]), but we did find seasonal changes: the smallest species, *Stolothrissa*, showed a significantly higher C:N ratio, i.e. lipid content, at the end of the productive dry season (Fig 6), which translates to >40% change in lipid content according to the model of Post et al. [56]. Seasonal lipid cycling is expected to be more pronounced in smaller fish, due to higher metabolic rates [95] and their planktivorous diet. While predators, such as *Lates stappersii* and large *Limnothrissa*, feed on both fish and zooplankton, the solely planktivorous *Stolothrissa* must cope with the strong seasonal fluctuations in plankton productivity. Thus, we might expect *Stolothrissa* to have a life history adapted to building reserves during the productive dry season for the following rainy season, when resources are less abundant. Alternatively, the changing C:N may relate to spawning activities [96]. However, spawning peaks were reported to occur in September and April-July [97, 98], i.e. during both our sampling occasions (Sep/Oct and Apr/May), and can thus not explain the observed changes in C:N between those two time points. The seasonal effect was less pronounced in the slightly larger *Limnothrissa* and was clearly absent in *Lates stappersii*, possibly due to their larger sizes and more piscivorous diets (Fig 9e).

Overall, the $\delta^{15}$N and C:N values indicate similar trophic levels and lipid contents of the northern and southern fish stocks. We hypothesize that the long term gene flow across the lake may inhibit the development of ecological differences among basins, despite the persistence of regional fish stocks at seasonal time scales.

## 5. Conclusions

In this study, we showed that the seasonal upwelling and mixing in the south basin of Lake Tanganyika induce distinct isotopic imprints at the primary producer level. These distinct isotopic signals can be tracked across the entire pelagic food web. Using $\delta^{13}$C as tracer, we identified fish stocks with regional foraging grounds, implying some degree of restricted movement on a seasonal and basin-wide scale. Correspondingly, regional fishery management strategies should consider including basin-scale quotas to maintain the food web structure in each basin. Our elemental and bulk isotopic composition data provide no clear evidence for strong physiological or dietary differences among these regional fish stocks. Although lake-wide gene flow may inhibit the evolution of clear ecological differences between regions/basins, different methods could yet reveal some ecological variation that cannot be resolved with bulk elemental and isotopic analyses. In the context of assessing the vulnerability of Lake Tanganyika's pelagic food web in a warming climate, our study indicates that the economically relevant pelagic fish species lack genetic structure indicative of local adaptation to basin-scale environmental differences, although they form regional stocks at seasonal time scales.

## Supporting information

**S1 Fig.** C:N mass ratio of (**a,b**) *Stolothrissa tanganicae*, (**c,d**) *Limnothrissa miodon*, and (**e,f**) *Lates stappersii* versus standard length in the northern and southern basins during the end of the dry season and the end of the rainy season. Only stations 1, 2 (north) and 7, 9 (south) are depicted. The shaded areas mark the 50 mm cut-off range for the population comparisons used in Fig 6.
(TIF)

**S2 Fig.** C:N corrected [56] $\delta^{13}$C of (**a,b**) *Stolothrissa tanganicae*, (**c,d**) *Limnothrissa miodon*, and (**e,f**) *Lates stappersii* versus standard length for the end of the dry season and the end of the rainy season. Only stations 1, 2 (north) and 7, 9 (south) are depicted. The shaded areas mark the 50 mm cut-off range for the population comparisons used in Fig 5. The sampled populations of *L. miodon* and *L. stappersii* from the end of the rainy season (d,f) were characterized by dense clusters of observations within a narrow size and $\delta^{13}$C range which may have skewed the basin-scale comparisons.
(TIF)

**S3 Fig. Distribution of ammonium (a) at the end of the dry season (Sep/Oct 2017) and (b) at the end of the rainy season (Apr/May 2018).**
(TIF)

**S4 Fig.** C:N corrected according to Post et al. [56] $\delta^{13}$C (left) and $\delta^{15}$N (right) of (**a,b**) *Stolothrissa tanganicae*, (**c,d**) *Limnothrissa miodon*, (**e,f**) *Lates stappersii*, (**g,h**) *Lates microlepis*, (**i,j**) *Lates mariae* and (**k,l**) *Lates angustifrons* versus standard length including all sampling locations and campaigns. Samples from the central basin and July 2017 were included for completeness, but were not included in the north-south and seasonal analysis presented in Fig 5. Note the different y-axis scaling.
(TIF)

**S5 Fig.** Carbon (normalized for C:N mass ratio according to Post et al. [56]) stable isotope signatures of *Stolothrissa tanganicae*, (**c,d**) *Limnothrissa miodon*, (**e,f**) *Lates stappersii*, including samples from the central basin, at the end of the dry season (left) and the end of the rainy season (right).
(TIF)

**S6 Fig.** Carbon (normalized for C:N mass ratio according to Post et al. [56]) and nitrogen stable isotope signatures of the large *Lates* species, namely (**a,b**) *Lates microlepis* (**c,d**) *Lates mariae*, and (**e,f**) *Lates angustifrons* at the end of the dry season (left) and the end of the rainy season (right). Orange dots represent the northern basin (stations 1–3) and blue dots represent the southern basin (stations 7–9). Numbers indicate the mean $\delta^{13}C$ of a population. Only individuals >150 mm were included in this analysis to reduce ontogenetic effects on the isotope signatures.
(TIF)

**S7 Fig. Lipid content versus C:N ratios of *Stolothrissa tanganicae*.** Each dot represents a tissue sample from one specimen.
(TIF)

**S8 Fig. Spearman-rank correlation matrix of physical, plankton, $\delta^{13}C$, $\delta^{15}N$ and C:N variables for the end of the dry season.** We selected the five stations across the north-south transect with the highest overlap among all variables (stations 1, 2, 6, 7, 9; S2 Table). Insignificant correlations ($p > 0.05$) are marked by grey crosses. Depth thermo: depth of the primary thermocline; $N^2$ thermo: buoyancy frequency of the primary thermocline; $Sc_{50-100\,m}$: Schmidt stability of the 50–100 m depth interval; $Phyto_{10\,\mu m}$: phytoplankton abundance of the >10 μm size fraction; $Zoo_{25/95/250\,\mu m}$: zooplankton parameters of the >25, >95, or >250 μm size fractions.
(TIF)

**S9 Fig. Spearman-rank correlation matrix of physical, plankton, $\delta^{13}C$, $\delta^{15}N$ and C:N variables for the end of the rainy season.** We selected the five stations across the north-south transect with the highest overlap among all variables (stations 1, 2, 4, 7, 8; S2 Table). Insignificant correlations ($p > 0.05$) are marked by grey crosses. Depth thermo: depth of the primary thermocline; $N^2$ thermo: buoyancy frequency of the primary thermocline; $Sc_{50-100\,m}$: Schmidt stability of the 50–100 m depth interval; $Phyto_{10\,\mu m}$: phytoplankton abundance of the >10 μm size fraction; $Zoo_{25/95/250\,\mu m}$: zooplankton parameters of the >25, >95, or >250 μm size fractions.
(TIF)

**S10 Fig. Experimentally determined $CO_2$ fixation rates in comparison to the oxygen and chlorophyll-*a* distributions at the end of the rainy season (Apr/May 2018).** (**a,c**) Oxygen and in-situ chlorophyll-*a* as well as (**b,d**) $CO_2$ fixation rates from stations 2 in the north (**a,b**) and 7 in the south (**c,d**).
(TIF)

**S11 Fig. Distribution and isotopic composition of dissolved inorganic carbon (DIC) at the end of the rainy season (Apr/May 2018) in Lake Tanganyika.** (**a**) DIC concentration profiles from stations 2, 5, and 7. (**b**) $\delta^{13}C$-DIC profiles from stations 1, 3, and 8.
(TIF)

**S12 Fig. Zooplankton community compositions per station for various net types (25 μm, 95 μm, 250 μm) during end of the dry season (top) and the end of the rainy season**

**(bottom).**
(TIF)

**S13 Fig. Nitrogen stable isotope signatures of different genetic clusters within the studied fish species in (left) July 2017, (middle) September/October 2017, and (right) April/May 2018.** (**a**-**c**) *Limnothrissa miodon*, (**d**-**f**) *Lates stappersii*, (**g**-**i**) *Lates microlepis* (**j**-**l**), *Lates mariae*, and (**m**-**o**) *Lates angustifrons*. Note the different axis scaling between *Limnothrissa* and the *Lates* species.
(TIF)

**S1 Table. Summary table compiling the physicochemical and biological variables for the end of the dry season (Sep/Oct 2017) and the end of the rainy season (Apr/May 2018).**
(XLSX)

**S2 Table. Data sets used for the correlation matrixes shown in Fig 6 as well as S10 and S11 Figs.** Of the chosen variables, chlorophyll-*a* and all POM-related parameters depict depth-integrated values. The $\delta^{13}$C, $\delta^{15}$N, and C:N values from all other food web members (except POM) represent average values from the respective sites. Gaps in the data set are highlighted in white. Gaps at the northern (station 1) or southern (station 9) extremities of the lake were filled by assuming the same value as from the neighbouring site. Other gaps were filled by calculating the average value between the two neighbouring sites. Rows (i.e. stations) used for calculating the correlation matrixes are highlighted in bold black font.
(XLSX)

# Acknowledgments

We are grateful for the support from our research collaborators at the Tanzania Fisheries Research Institute, particularly the Directors Rashid Tamatamah and Semvua Mzighani as well as Mary Kishe. Special thanks go to Mupape Mukuli as well as the captain and crew of the *M/V Maman Benita* for their steady toil in organizing and conducting the cruise work with us. We also thank Andreas Brand, Kathrin B.L. Baumann, and Tumaini M. Kamulali for their help during field work, Serge Robert and Fabian Kuhn for assistance in the lab, and Eliane Scharmin for administrative support. Special thanks go to Jessica A. Rick for providing help in the field, the data of the *Late*s genetic clusters, and comments on the manuscript. Thanks to Blake Matthews for insightful discussions. We furthermore thank the three anonymous reviewers for their helpful and constructive feedback. Thanks also go to the Tanzania Commission for Science and Technology (COSTECH) for granting the research permits.

# Author Contributions

**Conceptualization:** Benedikt Ehrenfels, Julian Junker, Ismael A. Kimirei, Ole Seehausen, Catherine E. Wagner, Bernhard Wehrli.

**Data curation:** Benedikt Ehrenfels, Julian Junker, Demmy Namutebi, Cameron M. Callbeck, Anthony Kalangali, Athanasio S. Mbonde.

**Formal analysis:** Benedikt Ehrenfels, Julian Junker, Demmy Namutebi, Cameron M. Callbeck.

**Funding acquisition:** Julian Junker, Ismael A. Kimirei, Carsten J. Schubert, Ole Seehausen, Catherine E. Wagner, Bernhard Wehrli.

**Investigation:** Benedikt Ehrenfels, Julian Junker, Cameron M. Callbeck, Christian Dinkel, Anthony Kalangali, Ismael A. Kimirei, Athanasio S. Mbonde, Julieth B. Mosille, Emmanuel A. Sweke.

**Methodology:** Benedikt Ehrenfels, Julian Junker, Cameron M. Callbeck, Christian Dinkel, Anthony Kalangali, Ismael A. Kimirei, Athanasio S. Mbonde, Julieth B. Mosille, Emmanuel A. Sweke, Carsten J. Schubert, Ole Seehausen, Catherine E. Wagner, Bernhard Wehrli.

**Project administration:** Benedikt Ehrenfels, Julian Junker, Christian Dinkel, Anthony Kalangali, Ismael A. Kimirei, Athanasio S. Mbonde, Julieth B. Mosille, Emmanuel A. Sweke, Ole Seehausen, Catherine E. Wagner, Bernhard Wehrli.

**Resources:** Ismael A. Kimirei, Emmanuel A. Sweke, Carsten J. Schubert, Ole Seehausen, Catherine E. Wagner, Bernhard Wehrli.

**Supervision:** Carsten J. Schubert, Ole Seehausen, Catherine E. Wagner, Bernhard Wehrli.

**Validation:** Benedikt Ehrenfels, Julian Junker, Demmy Namutebi, Cameron M. Callbeck, Christian Dinkel, Anthony Kalangali, Athanasio S. Mbonde, Julieth B. Mosille, Emmanuel A. Sweke.

**Visualization:** Benedikt Ehrenfels.

**Writing – original draft:** Benedikt Ehrenfels.

**Writing – review & editing:** Benedikt Ehrenfels, Julian Junker, Cameron M. Callbeck, Ismael A. Kimirei, Carsten J. Schubert, Ole Seehausen, Catherine E. Wagner, Bernhard Wehrli.

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
