## [Decision Letter · Decision Letter 0]

7 Apr 2022

PONE-D-22-02571Isotopic signatures induced by upwelling reveal regional fish populations in Lake TanganyikaPLOS ONE

Dear Dr. Ehrenfels,

Thank you for submitting your manuscript to PLOS ONE. After careful consideration, we feel that it has merit but does not fully meet PLOS ONE’s publication criteria as it currently stands. Therefore, we invite you to submit a revised version of the manuscript that addresses the points raised during the review process.

I have now received three reviews of your manuscript. Because I first received two very different reviews I decided to send your manuscript to a third reviewer. Two of the reviews give rather coherent judgements which are in line with my own assessment of your manuscript. These reviewers, as do I, point to the problem that the manuscript lacks a necessary synthesis analyses of the results to be able to see whether or not they support your conclusions. You present very many results on which your final rather few conclusions rest which is challenging regarding use of statistical approaches. One of the reviewers suggests more synthetic analyses using multistatistical approaches (PCA, NMDS) to be able to demonstrate the relative strength of different driving variables.  The other reviewer suggests to give your results a better structure and focus using tables and/supplement to store details. I agree with both suggestions. Your result section, as it currently stands, is not convincing in supporting your discussion and conclusions. This is symptomatic from that the whole study lacks an overall appropriate statistical approach for analyzing basin-scale dynamics of food webs. Thus, you need to supply better statistical arguments to convince the reviewers and me that there is good support for your conclusions

The language is not up to standards as pointed out by one of the reviewers, for example in the abstract. This also includes many errors regarding references to statements e.g. lines 113-115, 119-120 and proper descriptions of statistical analyses (degrees of freedom, R2-values of regressions etc.). Also, I did not find numbers of sample sizes and there are often statements on differences without any analyses given. (e.g. lines 371-373, 382-383, 391-393, the whole paragraph starting with line 412). You need to thoroughly go through the whole manuscript to correct these types of errors. The success of your manuscript clearly depends on how you handle the reviewers’ and my comments. Therefore, in your response letter you need to supply detailed point by point comments to how you have dealt with the reviewers' and my comments

We look forward to receiving your revised manuscript.

Kind regards,

Peter Eklöv

Academic Editor

PLOS ONE

Journal Requirements:

(We are grateful for the support from our research collaborators at the Tanzania Fisheries Research Institute, particularly the Directors Rashid Tamatamah and Semvua Mzighani as well as Mary Kishe. Special thanks go to Mupape Mukuli as well as the captain and crew of the M/V Maman Benita for their steady toil in organizing and conducting the cruise work with us. We also thank Andreas Brand, Kathrin B.L. Baumann, and Tumaini M. Kamulali for their help during field work, Serge Robert and Fabian Kuhn for assistance in the lab, and Eliane Scharmin for administrative support. Special thanks go to Jessica A. Rick for providing help in the field, the data of the Lates genetic clusters, and comments on the manuscript. Thanks to Blake Matthews for insightful discussions. This work was funded by the Swiss National Science Foundation (grant CR23I2-166589). Thanks to the Tanzania Commission for Science and Technology (COSTECH) for granting the research  permits.)

(This work was funded by the Swiss National Science Foundation. The grant CR23I2-166589 for the project titled “From biogeochemistry to the ecological genomics of pelagic fish stocks - a study across 4 trophic levels” was awarded to Bernhard Wehrli and Ole Seehausen (https://data.snf.ch/grants/grant/166589).

The funders had no role in study design, data collection and analysis, decision to publish, or preparation of the manuscript.)

Reviewers' comments:

Reviewer's Responses to Questions

**Comments to the Author**

1. Is the manuscript technically sound, and do the data support the conclusions?

Reviewer #1: Yes

Reviewer #2: Partly

Reviewer #3: Yes

2. Has the statistical analysis been performed appropriately and rigorously? 

Reviewer #1: Yes

Reviewer #2: No

Reviewer #3: Yes

3. Have the authors made all data underlying the findings in their manuscript fully available?

Reviewer #1: Yes

Reviewer #2: Yes

Reviewer #3: Yes

4. Is the manuscript presented in an intelligible fashion and written in standard English?

Reviewer #1: Yes

Reviewer #2: Yes

Reviewer #3: Yes

5. Review Comments to the Author

Reviewer #1: This manuscript concerns measurements of stable carbon and nitrogen isotopes in fish tissues from Lake Tanganyika. Seasonally varying hydrology, with respect to patterns of upwelling and nutrient availability are hypothesized to affect changes to fish trophic structure – and thus community ecological function. A suite of limnological and biogeochemical parameters were measured to contextualize isotope measurements. The statistical analysis was sound, however sample sizes for some ellipses were quite small. This value wasn’t interpreted heavily though. Overall, interpretation is sound.

Reviewer #2: This paper study patterns of stable isotopes across the food web in Lake Tanganyika when the whole lake is stratified and when water is circulating resulting in differences in stratification and upwelling between the north and south parts of the Lake. The results suggest that differences in isotopic signal traverse the food web when there is water circulation but not when the lake is stratified. One conclusion of this pattern is that pelagic fish use “regional grounds” for foraging that should be accounted for in management.

Although there are no direct flaws in the paper it is very hard to read. Mainly the result section is very long and difficult to grasp the important information and I suggest to put detailed numbers and differences into a Table or Supplement. The result section could almost be summerised in: at the end of the rainy season no differences and at the end of dry season there was differences in the isotoic signal that was tracked through the food web.

I also have some concern regarding the title and the conclusion about “regional fish population”. The fish populations pick up the dC signal of their food but that does not mean they are “regional” (this is a bit problematic in itself as it is not defined what is meant, maybe causing a confusion). As far as I understand the isotopic signal does not reveal anything about where they spawn or how they move in the lake. Of course they don’t swim back and forth on a weekly basis but I don’t see any evidence against that a fish is in one part of the lake in one period and 7 month later in the other part, it will just pick up the current regional isotope signal. It would be informative to know how, when and where these fish species spawn, pelagic, demersal, littoral spwners, buoyant eggs? Although I can find it likely with some kind of regional population structure of these fish species I cannot see the evidence against it.

Specific comments

Abstract

Poorly written with many odd sentences, e.g. l. 26 (global warmin related), l. 31 (differences in habitat), l. 36-38 (regional forage grounds and record…gradients), l. 39-40 (regional population on a seasonal…), l 41 those?, l. 42-43 Not part of this study at all, l. 45-46 you don’t show these at all, l. 48 how does basin relate to regional?

Methods:

Reference or motivation to statement on l. 258.

l. 295. Fish was sampled from landing sites, not the sampling stations.

l. 303. In general are mixing models used to infer the diet of consumers from stable isotopes, but that has not been applied here, or? If not, why? Wouldn’t this show if they differ in diet between northern and southern area just not isotope signal?

Results

Far too long and detailed, see above.

l. 435 Compare with l. 415-418. Don’t seem to match to me.

Discussion:

l. 569-571: Why were these fractions not analyzed, sampling issues?

l. 580-581, Can’t there be competiotion from the nano- and picc phytoplankton also?

l. 607-620 Feels reduntant in this context.

l. 702, “phenotypic changes” is a bit odd in this context. From this study there is no indication (not studied) fish make any phenotypic changes (as number of gill rakers or gut length) or behavior (vertical migration, diet changes), all changes may just be an effect of the change in isotope signal of prey.

Reviewer #3: The authors provide seasonal isotopic data of two sites of Lake Tanganyika to study the regionality of the fish population. Authors have carried out massive sampling and provide detailed analysis based on the carbon and nitrogen isotopes. My main suggestion to the authors is to provide a picture of the whole food web in Lake Tanganyika based on the literature (add this as figure 1). Moreover, authors compare dry and rainy season and it would be interesting to know how phytoplankton communities are assumed to differ based on the literature. The introduction would greatly benefit if the introduction could describe whole food web starting from the phytoplankton – zooplankton -fish including a description of the main species. One could assume that upwelling is an important occasion for diatoms and diatom-based food web.

In the result section authors could also provide picture(s) of carbon and nitrogen values of all studied food web components of both sites and seasons. This could help readers understand seasonality and site impact at the whole food web level and to understand if the regionality is only related to the specific fish species or can we see systematical differences at the different trophic levels between north and south and seasonal effect. Or could you put whole of your data (isotopes, biotic and abiotic measurements) to the multistatistical analysis (PCA, NMDS) and show what is actually happening on these two seasons and sites.

Regarding on the nitrogen isotopes I wonder if the authors are aware of how upwelling influence on the nitrogen cycle and the uptake of nitrogen by primary producers (ammonium or nitrate, Bartrons 2009: DOI: 10.5194/bgd-6-11479-2009). This could explain differences in nitrogen values.

I recommend authors to add water temperature to picture 1 at least assumed range.

In the methods you describe that you have used the Folch method for lipid extraction, however, the Folch method uses chloroform, methanol, and water in the proportions 8:4: 3, please check your reference. Secondly, you do not have supernatant in the lipid analysis, but lower phase which includes lipids, and this phase is usually transferred to the new tube.

In figure 3 you could provide letter on which site station is located e.g. station 1 (N) or station 8 (S).

In line 541 you say that lipid content reduced by 43 to 45%, however, I would keep it more informative if you could provide real values, e.g. lipid content reduced from x to y.

6. PLOS authors have the option to publish the peer review history of their article (what does this mean?). If published, this will include your full peer review and any attached files.

Reviewer #1: No

Reviewer #2: No

Reviewer #3: No

---

## [Author Response · Author response to Decision Letter 0]

29 Jul 2022

Below is a point-by-point response to all reviewer comments. Changes in the manuscript are documented in the mark-up mode version of the revised manuscript.

Reviewer #1: This manuscript concerns measurements of stable carbon and nitrogen isotopes in fish tissues from Lake Tanganyika. Seasonally varying hydrology, with respect to patterns of upwelling and nutrient availability are hypothesized to affect changes to fish trophic structure – and thus community ecological function. A suite of limnological and biogeochemical parameters were measured to contextualize isotope measurements. The statistical analysis was sound, however sample sizes for some ellipses were quite small. This value wasn’t interpreted heavily though. Overall, interpretation is sound.

REPLY: Agreed. Also considering the suggestions from the other reviewer and the editor, we analyzed our data set in a more comprehensive fashion. The key pieces are the two correlation matrixes for both seasons (see figures 7, S7, and S8). These correlations yielded consistent results that we think are useful in interpreting the key patterns emerging from the dataset. Even though not all relationships that we refer to were statistically significant, they were consistent, systematic, and unsurprising from a causal point of view. The test parameters are discussed in L.695-700: 

“Despite the congruent patterns across the food web, it is not surprising that most correlations were statistically insignificant, given our sample size of five and the chosen Spearman's rank method. Rank-based tests sacrifice explanatory power in favor of not assuming normal distribution.”

More information underlying all analyses, including the mean values, standard deviations, and sample sizes, are now compiled in supplementary tables (see S1 and S2 tables). Even though we agree that discussing some statistical parameters, such as sample size, in more detail would be worthwhile, we did not discuss those deeper in the manuscript. This decision grounds on two reasons:

1) Our analyses revealed significant north-south differences in �13C for all organisms with large sample sizes (3 ≤ n ≤ 74), i.e. P. spekii and the three major fish species (see figure 5) and clear, but not significant north-south differences for phyto- and zooplankton with lower sample sizes (n ≤ 3). We think that these systematic and coherent differences are sufficiently convincing for the qualitative conclusion that we draw: there is a significant north-south deviation in �13C across all trophic levels linked to the dry season upwelling/mixing. We do not draw quantitative conclusions about more extent of particular factors on the absolute isotope values, i.e. trying to find the true mean of a distribution, which would require a more in-depth statistical analysis.

2) In connection to the qualitative nature of our argumentation – and given the length and complexity of the manuscript – we prioritized discussing context that is important for understanding the causal links on which we found our argumentation (e.g. DIC dynamics, primary productivity rates, tissue turnover times). 

 

Reviewer #2: This paper study patterns of stable isotopes across the food web in Lake Tanganyika when the whole lake is stratified and when water is circulating resulting in differences in stratification and upwelling between the north and south parts of the Lake. The results suggest that differences in isotopic signal traverse the food web when there is water circulation but not when the lake is stratified. One conclusion of this pattern is that pelagic fish use “regional grounds” for foraging that should be accounted for in management.

Although there are no direct flaws in the paper it is very hard to read. Mainly the result section is very long and difficult to grasp the important information and I suggest to put detailed numbers and differences into a Table or Supplement. The result section could almost be summerised in: at the end of the rainy season no differences and at the end of dry season there was differences in the isotoic signal that was tracked through the food web.

REPLY: Done. We streamlined the entire manuscript according to this suggestion, with most revisions focusing on the results chapter. Most prominently, we rewrote and shortened the hydrodynamics and biogeochemistry sections (3.1) and moved auxiliary results including the respective figures (formerly figures 3 and 4) to the supplements, i.e. DIC concentration and isotopic composition, CO2 fixation rates, and the zooplankton community composition.

In addition, we compiled the data in a summary table and additional calculated a statistical test across the data set (see figures 7, S7, and S8 as well as tables S1 and S2). We believe that this analysis not only improves readability, but also provides stronger support for the conclusions we drew.

I also have some concern regarding the title and the conclusion about “regional fish population”. The fish populations pick up the dC signal of their food but that does not mean they are “regional” (this is a bit problematic in itself as it is not defined what is meant, maybe causing a confusion). As far as I understand the isotopic signal does not reveal anything about where they spawn or how they move in the lake. Of course they don’t swim back and forth on a weekly basis but I don’t see any evidence against that a fish is in one part of the lake in one period and 7 month later in the other part, it will just pick up the current regional isotope signal. It would be informative to know how, when and where these fish species spawn, pelagic, demersal, littoral spwners, buoyant eggs? Although I can find it likely with some kind of regional population structure of these fish species I cannot see the evidence against it.

REPLY: Correct. Our genetic results do provide evidence that there is lake-wide gene flow at sufficiently long time scales (i.e. there is no notable genetic structure at the lake basin scale). Thus, by “regional populations” we mean regional on a basin-wide and seasonal scale (see abstract and the conclusion from section 4.2 in the discussion: L.701-716). There are several life history mechanisms which could reconcile these findings. For example, there could be substantial movement during early life stages that would act to genetically homogenize populations, followed by relatively little movement at later life stages. 

We agree that information on spawning locations would provide relevant context. Although knowing spawning ground locations would be interesting, we do not think that the spawning grounds will affect our results due to the chosen size ranges, which exclude juvenile fish from our analysis (see section 2.9). In support of this argument, north-south differences in �13C during the dry season were consistent across all trophic level, whereas differences with respect to spawning ground preferences exist among the studied fish species. In brief, Stolothrissa and Lates stapppersii spawn offshore, whereas Limnothrissa and the three larger Lates species, L. microlepis, L. mariae, and L. angustifrons nearshore (see e.g. Coulter, 1970, 1991; Matthes, 1967).

Given the length and complexity of the manuscript, we therefore chose not to include this information in the manuscript.

Specific comments

Abstract

Poorly written with many odd sentences, e.g. l. 26 (global warmin related), l. 31 (differences in habitat), l. 36-38 (regional forage grounds and record…gradients), l. 39-40 (regional population on a seasonal…), l 41 those?, 

REPLY: We improved the phrasing throughout the abstract to make links clearer and achieve more coherence. 

l. 42-43 Not part of this study at all, 

REPLY: We integrated the genetic results from the previous publications within this project here. Drawing conclusions about the time span in which the fish populations can be regarded as “regional”, was only possible by combining the genetic results with the additional findings from the isotope data. Those two sets of data provide information on the distribution of the fish over long (genetics) or relatively short (isotopes) timescales. Additionally, we incorporated the existing genetic data into new ways here, i.e. we analyzed whether they correlate with the isotope data (see Supplementary Figure S11; formerly Fig. S7). 

l. 45-46 you don’t show these at all,

REPLY: See Supplementary Figure S11 (formerly Fig. S7). Nonetheless, we have removed the respective sentence from the abstract, because it was not a main result.

l. 48 how does basin relate to regional?

REPLY: Those are the same. We have replaced “basin-wide” or “basin-scale” with “regional” throughout the abstract to establish more coherence and avoid confusion.

Methods:

Reference or motivation to statement on l. 258.

REPLY: Collecting fresh fish samples via net hauls and doing all the biogeochemical measurements and experiments at the same time was logistically not feasible. Thus, the “fish team” collected fish samples from local fishermen and processed the samples on-board, while the water column samples were retrieved and processed. 

The information that local fishermen usually fish within a 20 km radius from their landing sites is expert knowledge gained from long-term interviews and monitoring efforts by local TAFIRI researchers. There is no published reference which we could cite in this context.

l. 295. Fish was sampled from landing sites, not the sampling stations.

REPLY: Correct. We changed “stations” to “sites” or “landing sites adjacent to station …”. For simplicity and coherence, we refer to all sampling sites as “station” thereafter. 

l. 303. In general are mixing models used to infer the diet of consumers from stable isotopes, but that has not been applied here, or? If not, why? Wouldn’t this show if they differ in diet between northern and southern area just not isotope signal?

REPLY: We chose not to apply isotopic mixing models for three reasons. First, the �15N of potential prey organisms – the bulk zooplankton community – varied by up to 6.7 ‰ within the basins and maximum values were similar to the highest fish �15N values. Second, the �15N values of individual zooplankton taxa can vary greatly, with reported values ranging from 0.1 to 5.9 ‰ in Lake Tanganyika. Third, exact trophic discrimination factors are not known for Lake Tanganyika, but appear to largely deviate from the norm. Together these factors impede setting up a mixing model to quantify the trophic position of the fish.

We rephrased L.725-738 accordingly to better convey these arguments in the manuscript. Thereafter, we chose to interpret only the absolute �15N values to a level, that the context of our data allows (L.739-749).

Results

Far too long and detailed, see above.

l. 435 Compare with l. 415-418. Don’t seem to match to me.

REPLY: According to the new analyses and figures (see figures 4, 7, S7, and S11, as well as tables S1 and S2), we rewrote the entire section, which we believe is more consistent now (see section 3.2). 

Discussion:

l. 569-571: Why were these fractions not analyzed, sampling issues?

REPLY: The focus of the overall research project was on the medium to large sized phytoplankton taxa, specifically filamentous, diazotrophic cyanobacteria. To obtain sufficient amounts of phytoplankton cells in this oligotrophic lake, we chose to concentrate up to 10 litres of lake water through a 10 µm plankton net. These microscopic samples were used to verify molecular cyanobacterial markers (phycocyanin and phycoerythrin), whereas chlorophyll-a served as a total phytoplankton community proxy. For further details, please see Ehrenfels et al. (2021), Front Environ Sci., 9:277 (https://www.frontiersin.org/article/10.3389/fenvs.2021.716765).

l. 580-581, Can’t there be competiotion from the nano- and picc phytoplankton also?

REPLY: Yes, indeed. We discuss this in L.569-578.

l. 607-620 Feels reduntant in this context.

REPLY: There are different mechanisms underlying the C and N isotopic composition of POM, respectively. Thus, we think that discussing the �15N-POM in the context of the hydrodynamic regime and associated N sources of the primary producers deserves a paragraph here.

l. 702, “phenotypic changes” is a bit odd in this context. From this study there is no indication (not studied) fish make any phenotypic changes (as number of gill rakers or gut length) or behavior (vertical migration, diet changes), all changes may just be an effect of the change in isotope signal of prey.

REPLY: We rephrased this sentence and now explicitly refer to the phenotypic parameters studied in this paper. The respective part reads now as follows: “Despite the absence of pronounced spatial genetic structure in either of the sardine (Junker et al., 2020; De Keyzer et al., 2019) or Lates species (Rick et al., 2021), phenotypic traits, such as diets and lipid contents, may vary between regional fish populations in response to regionally different environments…”.

 

Reviewer #3: The authors provide seasonal isotopic data of two sites of Lake Tanganyika to study the regionality of the fish population. Authors have carried out massive sampling and provide detailed analysis based on the carbon and nitrogen isotopes. My main suggestion to the authors is to provide a picture of the whole food web in Lake Tanganyika based on the literature (add this as figure 1). 

REPLY: Done.

Moreover, authors compare dry and rainy season and it would be interesting to know how phytoplankton communities are assumed to differ based on the literature. The introduction would greatly benefit if the introduction could describe whole food web starting from the phytoplankton – zooplankton -fish including a description of the main species. One could assume that upwelling is an important occasion for diatoms and diatom-based food web.

REPLY: Done. We agree that a more precise depiction of the food web (both in text and graphically) helps to better understand the paper. We have thus added a conceptual figure to the introduction (see figure 1), briefly describe the major trophic relationships (L.57-62), and have included a sentence regarding the phytoplankton community composition (L.107-109).

In the result section authors could also provide picture(s) of carbon and nitrogen values of all studied food web components of both sites and seasons. This could help readers understand seasonality and site impact at the whole food web level and to understand if the regionality is only related to the specific fish species or can we see systematical differences at the different trophic levels between north and south and seasonal effect. Or could you put whole of your data (isotopes, biotic and abiotic measurements) to the multistatistical analysis (PCA, NMDS) and show what is actually happening on these two seasons and sites.

REPLY: Done. We compiled the data in summary tables and performed a statistical analysis across the data set (see figures 7, S7, and S8 as well as tables S1 and S2). We believe that this analysis not only improves understandability, but also strengthens the conclusions we drew from the data.

Only five sites with a high overlap across all variables of the data for both of the sampling campaigns. With so few “replicates”, we chose not to do a PCA or NMDS, and instead calculated rank-based correlation matrixes (Fig 6 and S10 Fig). We added our statistical approach to the methods section “2.9 Data analysis”:

“To test to what extent the physicochemical and biological variables correlate, we chose the five sites with the highest possible overlap across all variables (Sep/Oct: stations 1, 2, 6, 7, and 9; Apr/May: stations 1, 2, 4, 7, 8). The data represent either depth-integrated values or averages per site. In the resulting data set were 24 gaps compared to a total of 280 data points (10 sites and 28 variables). Gaps at the northern (station 1) or southern (station 9) extremities of the lake were filled by assuming the same value as from the neighbouring site. Other gaps were filled by calculating the average value between the two neighbouring sites (S2 Table). We produced the correlation matrixes using the R package corrplot (70) and calculated the Spearman's rank correlation coefficient (some variables were not normally distributed; Shapiro-Wilk-Test, p < 0.05).”

The results are described in section 3.5 and picked up throughout the discussion again.

Regarding on the nitrogen isotopes I wonder if the authors are aware of how upwelling influence on the nitrogen cycle and the uptake of nitrogen by primary producers (ammonium or nitrate, Bartrons 2009: DOI: 10.5194/bgd-6-11479-2009). This could explain differences in nitrogen values.

REPLY: Correct, upwelling may alter the availability of different nitrogen sources, such as ammonium or nitrate, to primary producers and thus, affect their resulting �15N value.

In the manuscript, we only discuss nitrate and nitrogen fixation as possible new sources of nitrogen, because upwelling was too weak to supply deep-water ammonium into the productive surface waters during our study period. Ammonium concentrations were below limit of detection in the upper 100 m (or deeper) at all stations during both sampling campaigns. We added the figure to the supplements (S3 Fig) and state in the results section that “Ammonium concentrations were below limit of detection in the upper 100 m during both sampling campaigns” (L.361).

I recommend authors to add water temperature to picture 1 at least assumed range.

REPLY: Done.

In the methods you describe that you have used the Folch method for lipid extraction, however, the Folch method uses chloroform, methanol, and water in the proportions 8:4: 3, please check your reference. Secondly, you do not have supernatant in the lipid analysis, but lower phase which includes lipids, and this phase is usually transferred to the new tube.

REPLY: Done. We adapted the reference to “Chen, I. S., Shen, C. S. J., & Sheppard, A. J. (1981). Comparison of methylene chloride and chloroform for the extraction of fats from food products. Journal of the American Oil Chemists’ Society, 58(5), 599-601”. We furthermore clarified that the lower phase (and not the supernatant) were transferred to the new tube.

In figure 3 you could provide letter on which site station is located e.g. station 1 (N) or station 8 (S).

REPLY: Done. We followed through also for supplementary figure S5 (formerly figure 4).

In line 541 you say that lipid content reduced by 43 to 45%, however, I would keep it more informative if you could provide real values, e.g. lipid content reduced from x to y.

REPLY: Done.

References

Coulter, G. W.: Population changes within a group of fish species in Lake Tanganyika following their exploitation, J. Fish Biol., 2, 329–353, 1970.

Coulter, G. W.: Lake Tanganyika and its life., British Museum of Natural History, London and Oxford University Press, Oxford., 1991.

Junker, J., Rick, J. A., Mcintyre, P. B., Kimirei, I., Sweke, E. A., Mosille, J. B., Wehrli, B., Dinkel, C., Mwaiko, S., Seehausen, O. and Wagner, C. E.: Structural genomic variation leads to genetic differentiation in Lake Tanganyika’s sardines, Mol. Ecol., 29, 3277–3298, doi:10.1111/mec.15559, 2020.

De Keyzer, E. L. R., De Corte, Z., Van Steenberge, M., Raeymaekers, J. A. M., Calboli, F. C. F., Kmentová, N., N’Sibula Mulimbwa, T., Virgilio, M., Vangestel, C., Mulungula, P. M., Volckaert, F. A. M. and Vanhove, M. P. M.: First genomic study on Lake Tanganyika sprat Stolothrissa tanganicae: A lack of population structure calls for integrated management of this important fisheries target species, BMC Evol. Biol., 19(1), 1–15, doi:10.1186/s12862-018-1325-8, 2019.

Matthes, H.: Preliminary investigations into the biology of the Lake Tanganyika Clupeidae, Fish. Res. Bull. Zambia, 4, 39–45, 1967.

Rick, J. A., Junker, J., Kimirei, I. A., Sweke, E. A., Mosille, J. B., Dinkel, C., Mwaiko, S., Seehausen, O. and Wagner, C. E.: The Genetic Population Structure of Lake Tanganyika’s Lates Species Flock, an Endemic Radiation of Pelagic Top Predators, J. Hered., esab072, doi:10.1093/jhered/esab072, 2021.

---

## [Decision Letter · Decision Letter 1]

30 Aug 2022

PONE-D-22-02571R1Isotopic signatures induced by upwelling reveal regional fish populations in Lake TanganyikaPLOS ONE

Dear Dr. Ehrenfels,

Thank you for submitting your manuscript to PLOS ONE. After careful consideration, we feel that it has merit but does not fully meet PLOS ONE’s publication criteria as it currently stands. Therefore, we invite you to submit a revised version of the manuscript that addresses the points raised during the review process. I have now received comments on your manuscript from two of the former reviewers. Although both found that the manuscript had improved in the line of their comments they also found that the message still needs to be clarified (see especially the result section). Most importantly, you need to convince the reviewers and me that your conclusions regarding regional fish populations and your suggestions of regional fisheries management hold according to data. As this is a major point of the manuscript on which you base your conclusions this also needs to be supported with a clear definition and hypothesis of what is expected from a "regional" population vs. lack of regionality (see especially comments of reviewer 2). You also need to respond to reviewer 3's comment on using multivariate statistics and merging single isotope pictures into one figure to show how different sites and season influence isotope value and food web structure.  Please submit your revised manuscript by Oct 14 2022 11:59PM. If you will need more time than this to complete your revisions, please reply to this message or contact the journal office at plosone@plos.org. Please include the following items when submitting your revised manuscript:A rebuttal letter that responds to each point raised by the academic editor and reviewer(s). You should upload this letter as a separate file labeled 'Response to Reviewers'.A marked-up copy of your manuscript that highlights changes made to the original version. You should upload this as a separate file labeled 'Revised Manuscript with Track Changes'.An unmarked version of your revised paper without tracked changes. You should upload this as a separate file labeled 'Manuscript'.

We look forward to receiving your revised manuscript.

Kind regards,

Peter Eklöv

Academic Editor

PLOS ONE

Reviewers' comments:

Reviewer's Responses to Questions

**Comments to the Author**

1. If the authors have adequately addressed your comments raised in a previous round of review and you feel that this manuscript is now acceptable for publication, you may indicate that here to bypass the “Comments to the Author” section, enter your conflict of interest statement in the “Confidential to Editor” section, and submit your "Accept" recommendation.

Reviewer #2: (No Response)

Reviewer #3: (No Response)

2. Is the manuscript technically sound, and do the data support the conclusions?

Reviewer #2: Partly

Reviewer #3: Yes

3. Has the statistical analysis been performed appropriately and rigorously? 

Reviewer #2: No

Reviewer #3: No

4. Have the authors made all data underlying the findings in their manuscript fully available?

Reviewer #2: Yes

Reviewer #3: No

5. Is the manuscript presented in an intelligible fashion and written in standard English?

Reviewer #2: No

Reviewer #3: Yes

6. Review Comments to the Author

Reviewer #2: Although there have been several improvements of the manuscript I still find some weaknesses. In general, I still find especially the result section rather long and inconclusively written with a lot of detailed numbers that I think could be referred to figures (or Tables) instead. There is also a mix of presenting significant and non-significant results (compare l. 376, 391). I think for example you could remove or substantially shorten l. 342-353, 355-363, 384-386, 391-403, 433-448, 462-474, 482-486.

Maybe I was a bit unclear in my previous review or I misunderstand the interpretations of the result, but I cannot see that one of the main conclusions; “regional fish populations” hold based on this data. To expand my thoughts on this a bit. In the introduction it is referred to a study by Logan et al that “…isotopic study of fish and their surrounding food web along a geographical gradient can reveal regional population isolation if environmental differences among sites translate into divergent isotopic signatures of regional or local fish populations”. This is true but the cited study consider a global study comparing three different tuna-species over 16 years, which your study is not even close to. Importantly, they study GRADIENTS in isotopic signals whereas your paper focus on north and south samples, not a gradient, compare with Fig. 8 & 9 in Logan et al. In your study, if you assume all fish move completely random (i.e. no regional/spatial structure) fish will pick up signal of the food-web where it feeds, i.e. the differences in Fig. 4 are reflected in the fish in Fig. 5, or? Maybe I miss something important as I find it hard to pick out the important information in the result section, but I just don’t get how random movements will differ from movements with some “center of gravity”, unless you think fish traverse ~450 km (4°?) within a couple of weeks. Some clearly stated hypothesis of what is expected if there are “regional fish population” or not is necessary, especially as you don’t have extensive gradients as in Logan et al. Moreover, in the Conclusions it says “…regional fishery management strategies should consider including basin-scale quotas.” But what is basin-scale, from the profile in Fig. 2 there are no apparent “basins”? Based on your results, how many basins and where should the borders be according to you?

I find the study to be interesting also without any conclusions about fish population structure. However, I think the current focus on regional fish populations may be misleading.

Reviewer #3: Your revised version of the manuscript has improved especially illustration of figures 1-3 is now much better. You have added some correlation analysis, but you have skipped the main point for multivariate statistical analysis, which is to shorten and clarify the message of your manuscript. So, the point is to remove single isotope pictures (Fig. 4 c, d, g, h, I, j and Fig. 5) and have one stable isotope picture in PCA or NMDS plot which can clearly demonstrate how different site and season influence on the isotope value and food web structure. So, I still urge you to put all stable isotope data in PCA or NMDS and show your stable isotope data as one figure. See e.g. figure 4 in Ramos et al. 2009, who also published their paper in PlosOne doi:10.1371/journal.pone.0006236.

7. PLOS authors have the option to publish the peer review history of their article (what does this mean?). If published, this will include your full peer review and any attached files.

Reviewer #2: No

Reviewer #3: No

---

## [Author Response · Author response to Decision Letter 1]

26 Nov 2022

Dear Dr. Eklöv,

We are pleased to resubmit our revised manuscript (PONE -D-22-02571) for consideration by PLOS ONE, previously titled “Isotopic signatures induced by upwelling reveal regional fish populations in Lake Tanganyika”.

In the updated manuscript, we carefully addressed the points raised by you and the two reviewers. Specifically, we streamlined the result section according to the reviewer’s comments. With respect to the regional distribution of the pelagic fish, we used a more accurate terminology and changed the title accordingly. Even without drawing conclusions about the population structure, we are convinced that the practical implication of considering regional fishing quotas still holds in light of what we can safely infer from our data: the pelagic fish do not move across the lake on a seasonal scale or longer.

We also followed the comments from reviewer 3. Nonetheless, we do not understand the benefit of doing a PCA with only two dimensions (�13C and �15N) – as suggested. We have thus included the C:N ratio in the PCA analysis. In order to preserve transparency and readability we kept the conventional isotope biplots and added a figure of the PCA.

We thank you and the reviewers for the constructive and helpful feedback. We are convinced that these revisions improve the clarity and consistency of the manuscript.

Many thanks for your kind consideration of this manuscript.

Best regards,

Benedikt Ehrenfels (on behalf of the author team)

In the attached file “Response to Reviewers” is a point-by-point response to all reviewer comments. Changes in the manuscript are documented in the mark-up mode version of the revised manuscript.

---

## [Decision Letter · Decision Letter 2]

21 Dec 2022

PONE-D-22-02571R2Pelagic fish in Lake Tanganyika are regionally sessile but lack local adaptationPLOS ONE

Dear Dr. Ehrenfels,

Thank you for submitting your manuscript to PLOS ONE. After careful consideration, we feel that it has merit but does not fully meet PLOS ONE’s publication criteria as it currently stands. Therefore, we invite you to submit a revised version of the manuscript that addresses the points raised during the review process.

 I have received comments from one of the reviewers and both this reviewer and I are quite happy about how you have dealt with the reviewers' comments and we both think that the manuscript is much clearer now. Nevertheless, there are still some minor clarifications needed, pointed out by reviewer 2, that you might want to consider, but these should be relatively easy to address.

We look forward to receiving your revised manuscript.

Kind regards,

Peter Eklöv

Academic Editor

PLOS ONE

Journal Requirements:

Reviewers' comments:

Reviewer's Responses to Questions

**Comments to the Author**

1. If the authors have adequately addressed your comments raised in a previous round of review and you feel that this manuscript is now acceptable for publication, you may indicate that here to bypass the “Comments to the Author” section, enter your conflict of interest statement in the “Confidential to Editor” section, and submit your "Accept" recommendation.

Reviewer #2: (No Response)

2. Is the manuscript technically sound, and do the data support the conclusions?

Reviewer #2: Partly

3. Has the statistical analysis been performed appropriately and rigorously? 

Reviewer #2: Yes

4. Have the authors made all data underlying the findings in their manuscript fully available?

Reviewer #2: Yes

5. Is the manuscript presented in an intelligible fashion and written in standard English?

Reviewer #2: Yes

6. Review Comments to the Author

Reviewer #2: Overall I think the authors have improved the writing, terminology and structure of the manuscript. I have some smaller issues but I think they can easlily be addressed by the authors.

1. I don't think the change of title was very successfull, don't really know what motivated this but think the former title is more informative.

2. I'm skeptic to 'regional sessile' as it imply they are some how fixed to some substrate. Maybe something along regional forage grounds that you use on l. 39 or so is better. As I understand actual spawning araes are not identidied?

3. To that respect, could region and basins be defined? it seems like the southern basin is >7 degree south, or? but what is northern basin/stock? On the map there is a central basin, but there were no fish samples from this basin if I got it right. So should this be a basin/region on its own or part of souty/north stock?

4. On l. 40-41 you write "...fish reside in a region for a season or longer." You should clarify that you have studied seasonal variation, what happens in a longer run you don't know. To me it seems fully possible that they aggregate in the south during the productive upwelling but then disperse over the lake for the rest of the year. There are many examples of fish where different stocks (spawning units) mix in a productive area for foraging and then return to more "native" areas for spawning (salmon maybe being the most extreme example). So you don't know what the long term distribution look like.

5. l. 127, exchange effects for barriers

6. l. 754-756: It may be the other way around, as the environment may be rather homogenous with, similar prey (and predators?) there are only very weak barriers to gene flow.

7. PLOS authors have the option to publish the peer review history of their article (what does this mean?). If published, this will include your full peer review and any attached files.

Reviewer #2: No

---

## [Author Response · Author response to Decision Letter 2]

29 Jan 2023

The letter to the editor and the response to reviewers were uploaded as separate files.

---

## [Editor Report · Decision Letter 3]

1 Feb 2023

Isotopic signatures induced by upwelling reveal regional fish stocks in Lake Tanganyika

PONE-D-22-02571R3

Dear Dr. Ehrenfels,

We’re pleased to inform you that your manuscript has been judged scientifically suitable for publication and will be formally accepted for publication once it meets all outstanding technical requirements.

Kind regards,

Peter Eklöv

Academic Editor

PLOS ONE
---

## [Editor Report · Acceptance letter]

10 Mar 2023

PONE-D-22-02571R3 

Isotopic signatures induced by upwelling reveal regional fish stocks in Lake Tanganyika 

Dear Dr. Ehrenfels:

I'm pleased to inform you that your manuscript has been deemed suitable for publication in PLOS ONE. Congratulations! Your manuscript is now with our production department. 

Kind regards, 

on behalf of

Dr. Peter Eklöv 

Academic Editor

PLOS ONE